



# A modular field system enabling cavity ring-down spectroscopy of in-situ vapor observations in harsh environments: The ISE-CUBE system

Andrew W. Seidl[1], Harald Sodemann[1], and Hans Christian Steen-Larsen[1]

[1]Geophysical Institute and Bjerknes Centre for Climate Research, University of Bergen, Norway

**Correspondence:** Andrew W. Seidl (andrew.seidl@uib.no)

**Abstract.** Over the last two decades, cavity ring-down spectroscopy (CRDS) has allowed for increasingly widespread, in-situ observations of trace gases in vapor, including the stable isotopic composition of water vapor. However, in-situ observation in harsh environments pose a particular challenge, as these CRDS analyzers are designed for use in a conventional laboratory. As such, field deployments typically enclose the instrument in a "quasi-laboratory". These deployments often involve substantial

logistical effort, in addition to potentially affecting the measurement site, such as impacting flow conditions around near-surface processes. We designed the ISE-CUBE system as a modular CRDS deployment system for stable water isotope measurements, with a specific focus on observing near-surface processes. We tested the system during a two-week field campaign during Feb-March 2020 in Ny-Ålesund, Svalbard, Norway, with ambient temperatures down to $-30\,^{\circ}\mathrm{C}$, and winds gusting over $20\,\mathrm{m\,s^{-1}}$. The system functioned suitably throughout the campaign, with field periods exhibiting only a minimal decrease in isotopic

measurement precision ($\delta^{18}$O: $0.06\,‰$ & $\delta$D: $0.47\,‰$) as compared to optimal laboratory operation. Having proven itself in challenging arctic conditions, the ISE-CUBE system can be readily adapted to the particular needs of future stable water isotope researchers, wherever their research aims might take them.

## 1 Introduction

Understanding exchange processes between the atmosphere and surface is fundamental to constrain fluxes between com-

partments in the Earth System. There exist substantial knowledge gaps on the processes and their representation in models, especially at high latitudes and other cold environments. In these regions, strong gradients form in the surface layer during stable stratification, and govern fluxes of trace gases such as methane, carbon dioxide, and water vapor.

In particular, quantification of the evaporation and condensation flux of water vapor requires further in-situ measurements. Hereby, the stable isotope composition of the water vapor, as quantified by HD$^{16}$O (D=deuterium, $^2$H) and H$_2^{18}$O, as well as the

rarer H$_2^{17}$O, is a valuable asset in disentangling water vapor of different origin and undergoing different processes. For example, in 1961, Craig used the stable water isotope (SWI) composition of meteoric waters to establish the 8-to-1 linear relationship between H$_2^{18}$O and HD$^{16}$O. Dansgaard (1964) took this relationship further and detailed how it can encapsulate both latitudinal and continental information of the precipitation collection site. Since then, SWIs have been used to identify moisture source



origin (Kurita, 2011; Steen-Larsen et al., 2013), as well as to quantify mixing processes (Noone, 2012; Sodemann, 2017) in the
atmosphere. Many of these and other SWI studies have only been made possible through advances in observational technology.

Over the last two decades or so, laser spectroscopy has enabled the continuous, high-resolution observation of the SWI
composition in ambient air (Galewsky et al., 2016). In cavity ring-down spectroscopy (CRDS), the sample is guided through
a measurement cavity with highly precise pressure and temperature control, while measuring the decay of a laser pulse in
the infrared (Crosson et al., 2002; Gupta et al., 2009). Since the spectrometers are designed for set-up in a laboratory and
similarly controlled environments, the in-situ measurement of small-scale processes and gradients involve inlet lines with
manifolds (Steen-Larsen et al., 2013, 2014a; Berkelhammer et al., 2016), in addition to pre-existing structures (Bonne et al.,
2014; Galewsky et al., 2011) or tents (Steen-Larsen et al., 2013; Wahl et al., 2021). Munksgaard et al. (2011, 2012) applied
a pragmatic approach, whereby the analyzer and accompanying equipment were housed inside a single plastic chest (on the
order of $1\,\mathrm{m}^3$ in size) to facilitate their shipboard study of sea-water along the tropical northeastern Australian coast. A similar
approach has not yet been attempted for the in-situ measurement of water vapor isotopes in the cold environmental conditions
typical for high latitude winter.

Here we present a modular, in-situ CRDS measurement system, termed the ISE-CUBEs, that adequately protects the analyzer
from the harsh arctic environment, especially at extreme sub-zero temperatures. The ISE-CUBE system consists of a stack
of weather-proof plastic cases that are interconnected electrically and pneumatically for sample and ventilation air. With a
footprint of less than $1\,\mathrm{m}^2$ and a total volume of less than $0.5\,\mathrm{m}^3$, the ISE-CUBEs can be rapidly deployed at different sites
and is at least an order of magnitude smaller than conventional approaches. Attachment of a profiling sample arm allows
for the detailed measurement of surface layer gradients in the range of 0 to $2\,\mathrm{m}$ above the surface. An additional expansion
module allows for the collection of water vapor in a cold trap system for later laboratory analysis, including $\mathrm{H}_2^{17}\mathrm{O}$. After a
detailed description of the design principle and construction of the measurement system, we evaluate its performance and the
data quality based on measurements from a two-week field campaign at Ny-Ålesund, Svalbard, where the system encountered
strong winds and severe cold temperatures.

## 2 Measurement system design

The general aim of the ISE-CUBE system design was to enable ground-based in-situ measurements of the vapor isotope
composition without the need for additional sheltering. Installation and operation should be unaffected by weather condi-
tions, in particular wind and precipitation. Ambient deployment temperatures should have minimal effect on the instruments.
Wall effects in the inlet should be minimized, and condensation prevented, to avoid measurement artifacts from fractionation
and smoothed signals from memory effects. An inlet module should allow profiling of the surface layer with minimal flow
disturbance. The system should accommodate a cryogenic trapping module to collect discrete vapor samples for subsequent
laboratory analysis. Based on these requirements, we designed a modular system that in its core was based on waterproof plas-
tic containers, enabling setup by a single operator. We first give an overview of the overall arrangement of the measurement
system, before describing each module in more detail.





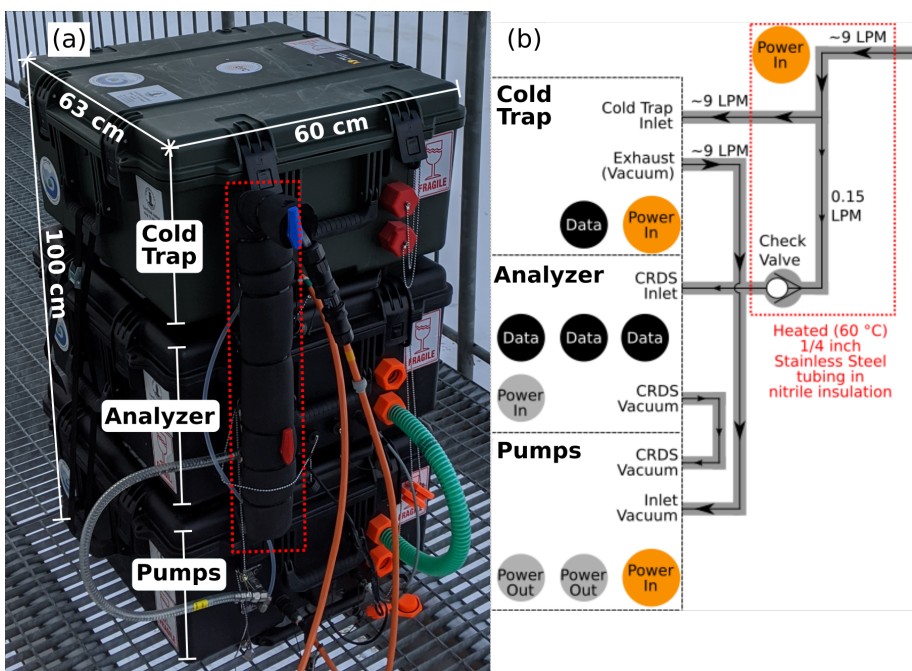

**Figure 1.** Overview of the ISE-CUBE system. (a) ISE-CUBEs in stacked configuration (from top to bottom): Cold Trap expansion module; Analyzer module; and Pump module. (b) Flow diagram for the entire ISE-CUBE system, including flow rates and connectors. A heated inlet assembly (red dotted area in both) pneumatically connects the three modules. See text for details.

## 2.1 Overall measurement system setup

The main modules of the ISE-CUBE system consist of a stack of the two primary modules (Analyzer and Pump modules), in addition to the Cold Trap expansion module; a list giving specifics of individual components can be found in Table A1 (and

even more details can be found in the Supplemental Material). All three modules use the same plastic container (iM2875 Storm, Pelican Products Inc), allowing for stacking and fastening of the the assembly, with a footprint of $0.38\,\mathrm{m}^2$ (see Figure 1). Each of the three modules is constantly ventilated with a $40\,\mathrm{m}^3\,\mathrm{h}^{-1}$ centrifugal fan. A plastic canvas can be placed over the stack for additional weather protection.

The stack is pneumatically interconnected via an inlet flow adapter (Figure 1, red dotted area) composed of approximately

$70\,\mathrm{cm}$ of $1/4\,\mathrm{inch}$ stainless steel tubing (Swagelok Inc.), heated to $60\,^\circ\mathrm{C}$ with self-regulating heat trace cabling (Thermon Inc.). A main flow of approximately $9\,\mathrm{L\,min}^{-1}$ is drawn into the adapter (Figure 1b) with most going through the Cold Trapping module, using a vacuum pump in the Pump module (Figure 1b, "Inlet vacuum"). Analyzer flow is split off prior to entering the Cold Trapping module, and a one-way check valve upstream of the analyzer inlet bulkhead (Figure 1b, "Check Valve") prevents reversal of the flow of sample air bound for the analyzer. The external vacuum pump of the CRDS analyzer (N920AP.29.18, KNF DAC GmbH) is also located in the Pump module (Figure 1b, "CRDS vacuum"). The inlet flow adapter also allows for

connection to additional lengths of inlet tubing, such as the Profiling module which enables sampling at adjustable heights (see



Sect. 2.5). We will now describe all four modules of the measurement system individually, beginning with the the Analyzer and Pump as the core modules.

## 2.2 Analyzer module

We use a Picarro CRDS water isotope analyzer (L2130-i, Picarro Inc.) as the central element of the Analyzer module (Figure 1, middle container). The Analyzer module is lined with custom-fit Low Density Polyethylene (LDPE) foam padding to protect the analyzer. Some padding can be removed to increase ventilation flow for better temperature regulation, depending on ambient conditions during field operations. With a power draw of $\sim 100\,\mathrm{W}$ in steady state there is substantial heating from the analyzer. Therefore, adequate ventilation is required to keep the analyzer internal temperatures ($T_{das}$) below about $50\,°\mathrm{C}$, as prolonged

exposure to higher temperatures (above $70\,°\mathrm{C}$) can permanently damage electrical components

The analyzer computer can be controlled from an external laptop through a ethernet (RJ45) cable, or via USB-connected monitor and keyboard/mouse combination. The particular analyzer used here (Ser#: HIDS2254) is a custom modification of the standard L2130-i, now enabled for high flow rates, similar to the analyzer described in Sodemann et al. (2017). The high flow rate is obtained by replacing the internal, constricting orifice ($70\,\mathrm{microns}$) needed for low-flow mode (typical flow rates

$0.035\,\mathrm{L\,min^{-1}}$) with a standard $1/4\,\mathrm{inch}$ stainless steel section. High flow rates (about $0.15\,\mathrm{L\,min^{-1}}$) enable faster analyzer response, but also cause more variable pressure inside the measurement cavity. The full impact of this particular configuration will be discussed in more detail below (Sect. 4.1). Sample air is guided from the exterior inlet bulkhead of the module into the analyzer via a $20\,\mathrm{cm}$ piece of flexible, $1/4\,\mathrm{inch}$ Polytetrafluoroethylene (PTFE) tubing. A similar length of $3/8\,\mathrm{inch}$, wire-reinforced, PVC tubing (K7160-06, Kuriyama of America Inc.) connects the vacuum port of the analyzer to the exterior

bulkhead. The vacuum and electrical bulkheads of the Analyzer module then connect to the Pump module.

## 2.3 Pump module

The Pump module contains the pumps and additional components necessary for operation of the analyzer (Figure 1, bottom container). The Pump module is connected to the Analyzer module with a $3/8\,\mathrm{inch}$, wire-reinforced, PVC tubing (K7160-06, Kuriyama of America Inc.), providing the vacuum necessary for the analyzer (Figure 1b, "CRDS Vacuum"). The Inlet vacuum

pump (N022AN.18, KNF DAC GmbH) continuously flushes the inlet tubing, providing the main flow for both the analyzer and the Cold Trapping module (see Sect. 2.4). Additionally, a small $300\,\mathrm{W}$ Uninterruptible Power Supply (UPS) (EL500FR, Eaton) inside the module protects the analyzer from possible power fluctuations and short power breaks. All components in the Pump module are strapped to an aluminum support frame, which itself is firmly wedged between lid and bottom of the case, when the case is shut. Thereby, we could avoid drilling unnecessary mounting holes in the plastic case. Together, the Analyzer

Module and the Pump Module are the two essential modules for in-situ isotopic analysis of water vapor, and allow stand-alone field operation. The Cold Trapping module and the Profiling module both expand upon the functionality of these core modules, as will be described next.





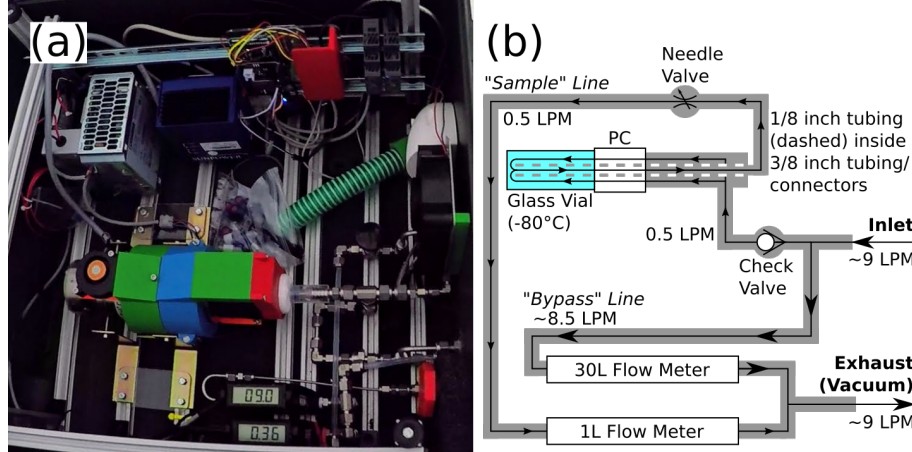

**Figure 2.** The Cold Trapping Module. (a) Interior of the module; cryocooler is surrounded by green, blue, and red plastic near photo center. (b) Flow diagram for the module. "PC" indicates placement of polycarbonate vial adapter. See text for details.

## 2.4 Cold Trapping expansion module

We included a cold trap module into the ISE-CUBE assembly, providing the ability to retrieve sample material from the field

for subsequent laboratory analysis. This discrete sample analysis could also allow for calibration of the measurements if no external vapor source is available. In addition, more precise analyses, as are needed for $^{17}$O, can be performed in the laboratory afterwards. Many commercially available Cold Trapping options involve the use of liquid cooling agents, such as ethanol or isopropyl alcohol. Peters and Yakir (2010) demonstrated the feasibility of collecting vapor samples with a Stirling cycle cryocooler as the cryogenic source of a cold trap. Due to the safe transportation and fast installation of such a cold trap, we

adopted the basic design of Peters and Yakir (2010) for the ISE-CUBE cold trap expansion module.

The cryocooler migrates heat away from the tip of a cryogenic "finger" towards the body of the cryocooler, where the heat is radiated away. By attaching this cryogenic finger to a thermally conducting mass (150 g of brass and aluminum) encircling a glass sampling vial, it directly cools the vial down to $-80\,°C$. Incoming water vapor in sample air (Figure 2b, "Inlet") is routed through a combination of $3/8$ inch stainless steel tubing and connectors, and is then introduced to the cooled glass collection

vial, connected to the large bore tubing with a custom-made polycarbonate adapter ("PC" in Figure 2b). Upon entering the vial, the water vapor is rapidly cooled below its frost point, and it collects on the interior walls of the glass vial. The dried air exits the glass vial via a $1/8$ inch length of stainless steel tubing, leading out of the end of the $3/8$ inch tubing/connector combination. Finally, this $1/8$ inch tubing is connected to the Inlet vacuum pump (Figures 1b & 2b, "Exhaust (Vacuum)"), which provides the necessary flow for the Cold Trap. The frozen sample is then sealed after the end of a sampling period, and

stored until laboratory analysis from the same vial. Also, the relatively small size of the setup easily fits within the standard ISE-CUBE container (Figure 1, top box).





We modified the original design of Peters and Yakir (2010) with regard to two aspects, namely the choice of cryocooler, and the flow configuration. Firstly, we opted for a more powerful cryocooler (L. Peters, personal communication, 20 March 2019), enabling faster and more consistent cooling of the sampling vial. Our chosen cryocooler (Figure 2a, underneath red, green, and blue plastic casing, Cryotel MT, Sunpower Inc.) takes 5 to 6 minutes to reach $-80\,°C$, at which temperature it has $23\,W$ of cooling power. Secondly, as the system was designed to work in concert with the Analyzer and Pump modules, we utilized the Inlet pump described in Sect. 2.3 to provide flow through the collection vial. Accomplishing this required the splitting of flow inside the Cold Trapping module into a "sample" and "bypass" line (Figure 2b). The "sample" line allowed incoming, moist air into the glass collection vial, with flow regulation (approximately $0.5\,L\,min^{-1}$ or less) via a manual needle valve. The "bypass" line carried the excess flow (approximately $8.5\,L\,min^{-1}$), ensuring that the flushing of the inlet was maintained. Flow rates through the "sample" and "bypass" lines were monitored using $1\,L\,min^{-1}$ and $30\,L\,min^{-1}$ mass flow meters (TopTrak 822, Sierra Instruments Inc.), respectively. These flow meters recorded onto an SD card via microcontroller (Mega, Arduino), which in turn allowed for remote monitoring of the flow rates via the external USB bulkhead. Splitting the lines in such a way allowed the Inlet pump to provide flow through both the Cold Trap collection vial and the inlet tubing, which included the Profiling expansion module.

## 2.5 Profiling expansion module

To enable detailed investigation of near-surface gradients of water vapor, and exchange processes between surface and atmosphere, we designed a profiling expansion module for the ISE-CUBE system. The Profiling module allows for the acquisition of vertical profiles of the ambient vapor at any height between 4 to $205\,cm$ above ground (Figure 3). Thereby, the lean design is expected to causes minimal flow disturbance at the measurement site. The module attaches to the inlet adapter and consists of approximately $4\,m$ of $1/4\,inch$ stainless steel tubing (Swagelok Inc.), heated to $60\,°C$ with self-regulating heat trace cable (Thermon Inc.), and surrounded with $2\,cm$ thick foam nitrile insulation. Profiling capabilities are enabled by an encasing of the final $1.9\,m$ of tubing by an aluminum articulating arm (Figure 3). The base of this arm is then attached to an aluminum mast and tripod. The tripod serves as the frame for a winch and pulley system to manually control the sampling height (Figure 3). Additional environmental parameters are acquired during profiling from a sensor package mounted on the "head" of the arm, near the air inlet. Height of the inlet is monitored by ultrasonic distance sensors (HC-SR04, SparkFun Elec.; Figure 3, red box). Air temperature at sampling height is measured using a temperature probe (VMA324, Velleman; blue rectangle in Figure 3b). Both variables are logged onto an SD card via microcontroller (UNO, Arduino) housed inside a weatherproof container (Figure 3a, yellow case at lower left).

While the descriptions of these two expansion modules are included here for completeness, only first-order assessments will ultimately be provided, as in-depth evaluation of the ISE-CUBE system is primarily focused on the performance of the analyzer in the Analyzer module (Sect. 4).





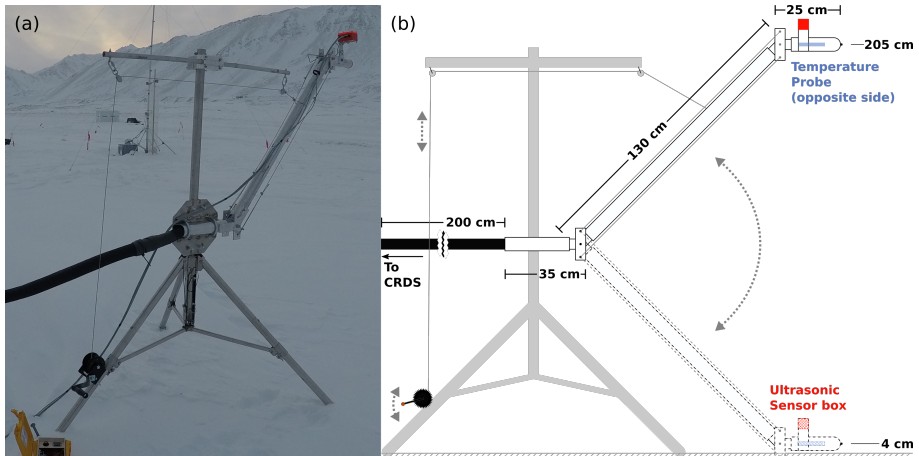

**Figure 3.** The Profiling module, with articulating arm. (a) Profiling module during field deployment. Inlet "head" in upper right of photo, with ultrasonic distance sensors encased in red plastic housing. Temperature sensor (not visible) located on opposite side of "head". Black winch in bottom left of photo controls inlet height via pulley system. Also visible in bottom left is the yellow case containing power supply and datalogger for temperature and distance sensors. Black tubing leading off to left connects to ISE-CUBE inlet. (b) Dimensional diagram for articulating arm.

## 2.6 Field calibration expansion module (proposed)

Isotopic measurements obtained in the field require proper calibration before they can be interpreted. The calibration procedure
involves providing the analyzer with a steady stream of vapor of known isotopic composition. These periods are subsequently
used to calibrate the raw measurements on the international VSMOW2-SLAP2 (Vienna Standard Mean Ocean Water 2 -
Standard Light Antarctic Precipitation 2) scale. It was our original intent to integrate a calibration module in the ISE-CUBE
system, allowing for in-field daily calibrations. However, our field calibration system was not ready in time for our measurement
campaign. Therefore, we calibrated the analyzer in a laboratory setting near to the deployment site immediately before and
after deployment at each measurement site (see Sect. 3.3). Nevertheless, such a calibration module has the potential to be
included in subsequent iterations of the ISE-CUBE system (see Sect. 5).

## 3 Performance test data sets

## 3.1 Campaign site and weather conditions

The performance of the ICE-CUBE system was evaluated based on a field campaign data set obtained in challenging Arctic
measurement conditions at the scientific settlement of Ny-Ålesund, Svalbard (Figure B1), during the ISLAS2020 campaign.
The principal aim of the ISLAS2020 field experiment was to obtain detailed in-situ measurements of isotope fractionation in
a cold, dry Arctic environment. Ny-Ålesund, with the adjacent strong air-sea interaction in the Fram Strait, is a well-suited





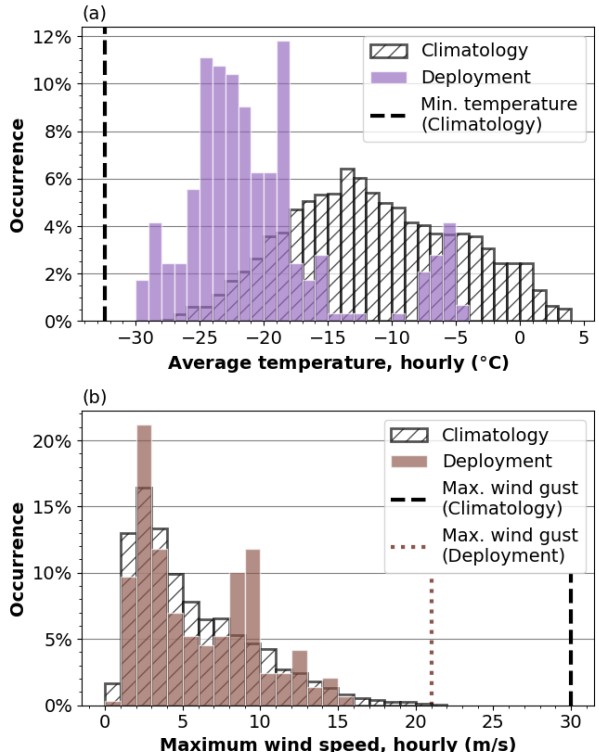

**Figure 4.** Climatologies of temperature and wind in Ny-Ålesund. (a) Histogram of hourly average temperatures for 2000–2019, 21 Feb to 14 Mar "Climatology" (black rectangles), alongside same from during the ISLAS2020 field campaign "Deployment" (purple bars). Black dashed line denotes minimum hourly temperature from the 2000–2019 period. (b) Same as (a), but for hourly maximum wind speeds, with brown bars denoting "Deployment" conditions. Lines indicate maximum hourly wind gusts over climatology (black dashed) and deployment (brown dotted) periods. From *Norsk Klimaservicesenter, Ny-Ålesund (SN99910)*.

location to make such observations. In particular during winter and spring, this region is frequently subject to periods of strong marine cold-air outbreaks, associated with intense evaporation (Papritz and Spengler, 2017).

In Ny-Ålesund, we deployed at two measurement sites, the first being approximately $300\,\mathrm{m}$ south of Ny-Ålesund, on the tundra ($78.921\,17\,°\mathrm{N}, 11.913\,61\,°\mathrm{E}$). This site was also referred to as the "Snow" location and was used from 25–28 Feb 2020. The second site was located on a concrete pier at the northernmost edge of the settlement (referred to as "Fjord"; $78.928\,73\,°\mathrm{N}, 11.935\,52\,°\mathrm{E}$), and was used from 7–14 March 2020.

General $2\,\mathrm{m}$ air temperature and $10\,\mathrm{m}$ wind speed and gust information for the Ny-Ålesund weather station (SN99910)
were retrieved from the Norsk Klimaservicesenter (https://seklima.met.no/observations). From this station dataset, $20\,\mathrm{year}$ climatological conditions were established by considering the hourly dataset of the period between 21 Feb and 14 Mar, between 2000 to 2019. The time period from 21 Feb to 14 Mar in Ny-Ålesund has climatological median air temperature for 2000–



2019 of $-12\,°C$, with $50\,\%$ of hourly average temperatures being between $-16\,°C$ and $-6\,°C$ (Figure 4a, black rectangles). In comparison, the majority of the ISLAS2020 campaign was spent in temperatures spanning $-26$ to $-18\,°C$, with a median value of $-20\,°C$ (Figure 4a, purple bars). The coldest temperatures experienced during the campaign ($-30\,°C$) were comparable to the minimum temperature in the 2000–2019 climatology, $-32.5\,°C$ (Figure 4a, black dashed line).

Climatological wind speed frequency peaks in the 2 to $3\,m\,s^{-1}$ range (Figure 4b, black rectangles). Hourly maximum wind speeds during the field deployment (Figure 4b, brown bars) peak in the same range, though winds had a secondary peak in the 8 to $10\,m\,s^{-1}$ range, twice the climatological frequency. With this distribution, we can classify winds during our observational period as mostly typical for the settlement. However, maximum wind gusts were still exceeding $20\,m\,s^{-1}$ at times during the campaign (Figure 4b, brown dotted line). Overall, the deployment conditions presented a useful test case for the system. In addition to cold temperatures, the combination of high winds, low temperatures, and precipitation provide a reliable situation to evaluate if the system was still able to acquire stable and reliable measurement data.

## 3.2 Data processing

The ISE-CUBEs produces three data streams, pertaining to the Analyzer, Cold Trap, and Profiling modules. An overview of the information contained in these datasets are listed in Table 1. The isotopic dataset generated by the analyzer in the Analyzer module is the primary dataset. The analyzer has been configured to measure at $4\,Hz$ frequency, with the primary environmental parameters observed being humidity (volumetric mixing ratio), $\delta^{18}O$, and $\delta D$, alongside a multitude of analyzer diagnostic metrics. The dataset from the Analyzer module is generated and stored by the internal computer of the CRDS analyzer. Before further use, the isotope data set was calibrated and corrected as described in Sec. 3.3 below.

**Table 1.** Parameters logged by the ISE-CUBE system. '*' indicates parameters classified as metadata, providing information on the quality of the data collected.

| Module | Parameter |
| --- | --- |
| Analyzer | Isotopes ($\delta^{18}O$, $\delta D$), |
| | Humidity, |
| | analyzer diagnostics* |
| | (temperatures, presures, etc.) |
| Cold Trap | Flows, |
| | Module temperature*, |
| | Cryocooler temperature range* [FLAG] |
| Profiling | Inlet height, |
| | Inlet temperature |

Both the Cold Trap and Profiling expansion modules measure at $1\,Hz$ via Arduino microprocessors, and record parameters to SD cards. The Cold Trap module monitors and records the flow through the collection vial, alongside remaining flow through



the "bypass" line, as well as the temperature inside the module container. The same module also flags whether the cryogenic finger is within $5\,\mathrm{K}$ of the target temperature ($-80\,^\circ\mathrm{C}$).

The Profiling module records the temperature at the inlet head, in addition to height distances measured by the ultrasonic sensors. Laboratory calibration of the temperature probe showed a systematic offset of $0.67\,\mathrm{K}$, which is accounted for during post-processing. Throughout the campaign, all module timestamps were compared to universal coordinated time, with offsets accounted for and datasets synchronised during processing.

  A python script package (IsoFuse) then averaged each of the three data streams along a common time vector, and joined
them into a netCDF file. While the technical analysis that will follow uses $1\,\mathrm{s}$ averaging with raw isotopic measurements, our final calibrated isotopic dataset is averaged over $30\,\mathrm{s}$ periods (see Sect. 4.5).

### 3.3 Isotope calibration and data processing

Calibrations of the analyzer were performed immediately preceding and following deployment at each measurement site, at the Marine Laboratory in Ny-Ålesund. Two secondary standards, calibrated on the VSMOW2-SLAP2 scale at FARLAB,
University of Bergen, Norway, were employed for calibration, namely DI ($\delta^{18}$O: $-7.68\,‰$ and $\delta$D: $-49.71\,‰$) and GSM1 ($\delta^{18}$O: $-32.90\,‰$ and $\delta$D: $-261.58\,‰$). Standards were delivered with the Standard Delivery Module (SDM) device (A0101, Picarro Inc.), utilizing a Drierite filled moisture trap as a source of dry air. Only sufficiently stable calibration signals lasting longer than $10\,\mathrm{minutes}$ are considered for use in calibrating the dataset. As mixing ratios at the deployment site can be well below $6000\,\mathrm{ppmv}$ for the time of year, a correction of the isotope composition was applied during post-processing, according
to the quantified mixing ratio – isotope ratio dependency by Weng et al. (2020), specific to the analyzer used in the field. Several calibrations made during the ISLAS2020 field deployment were between mixing ratios of $6000\,\mathrm{ppmv}$ and $850\,\mathrm{ppmv}$, and confirm overall agreement with the mixing ratio – isotope ratio dependency determined in the laboratory.

  Across 17 calibrations, DI measurements had a standard deviation of $0.15\,‰$ for $\delta^{18}$O, and $0.48\,‰$ for $\delta$D. These are similar (or smaller) values for the standard deviation typical of the individual calibrations. Total measurement drift across the
campaign duration for DI was found to be smaller than these standard deviations. The GSM1 standard experienced technical issues with the SDM during multiple calibrations. Only 17 of the 28 calibrations passed automatic quality control thresholds, such as humidity variation below $500\,\mathrm{ppmv}$. The standard deviation across the 17 valid calibrations was $0.18\,‰$ ($\delta^{18}$O) and $0.47\,‰$ ($\delta$D), again of a similar magnitude to the variabilities seen in individual calibrations. And as was also the case for DI, the total measurement drift across the campaign duration for GSM1 was found to be similar to or smaller than these standard
deviations. Overall, these drift values are compatible with the behavior exhibited by the same analyzer during previous use in the lab and field (Weng et al., 2020; Chazette et al., 2021) and exceed the manufacturers typical performance specifications (Picarro Inc., 2021).

### 3.4 Laboratory reference data

The CRDS analyzer contained in the Analyzer module is designed for use in well-controlled laboratory conditions. Therefore,
we define a reference time period when the same analyzer was used at FARLAB, University of Bergen, Norway. We use





the two month period of June–July 2020 as a reference benchmark with regard to instrument and cavity temperature and pressure variations, which could influence the spectroscopy of the analyzer. During this time, the analyzer was in use for routine measurement operations with ambient room temperatures of approximately $20\,°C$. This routine measurement included sampling mixing ratios down to $250\,ppmv$, comparable to humidities encountered in the field.

## 3.5   Additional datasets

Since 2010 the Alfred Wegner Institute (AWI) has operated an eddy-covariance (EC) station (Ny-Ålesund eddy-covariance site) (Jocher et al., 2012; Schulz, 2017), about $300\,m$ south of the settlement and approximately $20\,m$ away from the ISE-CUBE system during the initial deployment phase at Snow. This EC station measures, amongst others, air temperature and winds at four heights: $0.5\,m$, $1.0\,m$, $1.5\,m$, and $2.5\,m$, with half-minutely resolution available. Temperatures are measured at all levels by Thies Clima compact temperature sensors (2.1280.00.160, Adolf Thies GmbH & Co. KG), inside ventilated shields (1.1025.55.100, Adolf Thies GmbH & Co. KG). Wind data are from Young propeller-style wind monitors (5106-5 (5103-5 @ $2.5\,m$), R. M. Young Company). Temperature and winds at $0.5\,m$ were taken to be representative of ambient conditions while the ISE-CUBE system was deployed nearby, as detailed in Sect. 4.1. Additionally, temperature at $1.0\,m$ was used for field validation of the temperature probe fixed to the head of the Profiling module, as described in Sect. 4.4.

A temporary EC station was also installed approximately $13\,m$ away from the ISE-CUBE stack during the final phase of the campaign at the Fjord location. This setup included a Campbell Scientific CSAT3 sonic anemometer installed at $1.95\,m$ above ground, providing $20\,Hz$ measurement of wind and temperature. Minutely averages were derived from this high-resolution series, with less than $50\,\%$ low-quality data (as determined by internal CSAT3 diagnostics) over the minute. These measured conditions were taken to be representative of the ambient environment while the ISE-CUBE system was deployed at Fjord (Sect. 4.1).

## 4   Results

Now we detail how the field conditions influenced analyzer performance and thus data quality, using the laboratory performance as a benchmark. Thereby, we focus first on temperature and pressure conditions of the analyzer, before evaluating the impact of field conditions on the water isotope measurements. Finally, the performance of the cold trap and the profiling modules are briefly presented.

### 4.1   CRDS analyzer response to ambient conditions

Using our laboratory benchmark, we will detail how the field conditions influenced analyzer performance and thus data quality. We first compare the overall temperature of the analyzer using the Data Aquistion System (DAS) temperature ($T_{DAS}$) as a proxy (Picarro Inc., 2013). Then, we investigate the measurement cavity through its temperature ($T_C$) and pressure ($p_C$). Finally, we study the essential analyzer electronics (Wavelength Monitor (WLM)) via the warm box temperature ($T_{WB}$).



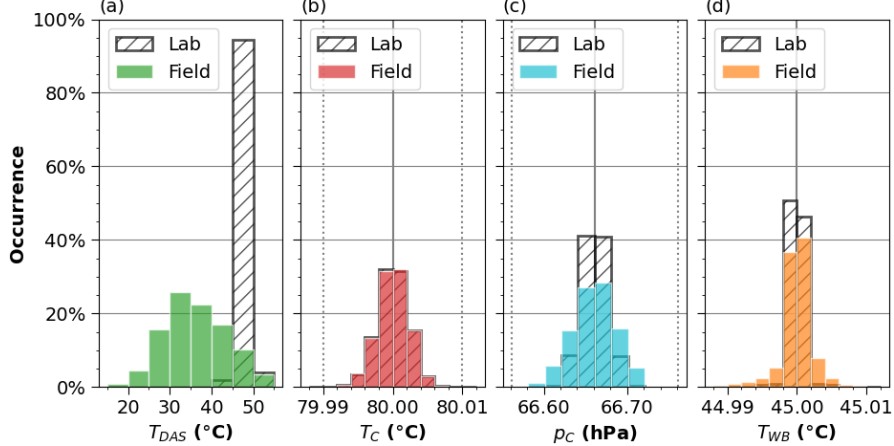

**Figure 5.** Relative occurrence of a) DAS temperatures, $T_{DAS}$ (5 °C bins), b) Cavity temperatures, $T_C$ (0.002 °C bins), c) Cavity pressures, $p_C$ (0.02 hPa bins), and d) Warm Box temperatures, $T_{WB}$ (0.002 °C bins) during CRDS operation, for field deployment periods (colored bars) as compared to laboratory benchmark (hatched bars). Solid vertical lines for b), c), and d) indicate parameter target value. Dotted lines for b) and c) indicate instrument control precision as per manufacturer.

### 4.1.1 Overall analyzer temperature

The $T_{DAS}$ serves as a first-order proxy for the overall measurement environment of the analyzer. As the $T_{DAS}$ results from a balance between radiant, excess heat from other components and constant ventilation with ambient air, its value can span a wide range. In the laboratory, $T_{DAS}$ values are typically within a narrow distribution, with 94.5 % of $T_{DAS}$ falling within the 45 to 50 °C bin (Figure 5a, hatched bars). In contrast, the most frequent $T_{DAS}$ from the field is contained within the 30 to 35 °C bin (25.9 %), and the overall distribution is markedly broader (Figure 5a, green bars). The median $T_{DAS}$ is 11.9 °C lower in field, while Interquartile Range (IQR) is 4.5 times broader (Table 2). Thus, the overall temperature of the analyzer was colder and more variable in the field as compared to the lab. We now investigate the possible origin of these differences in $T_{DAS}$.

Since the $T_{DAS}$ partly results from ambient air ventilated into the analyzer, we now investigate how ambient conditions relate to $T_{DAS}$. Analyzer $T_{DAS}$ medians were between 28 to 46 °C, whereas ambient temperatures varied between −29 to −5 °C during the 12 different deployment days at the Snow and Fjord site (Figure 6). Considering both measurement sites, the median daily $T_{DAS}$ (solid green line) exhibits a weak correlation (0.36) with daily mean temperature (Figure 6a, purple line). As there should be some degree of connectivity between analyzer temperature and the temperature of environmental air being ventilated into the module, this relation fits with expectation. However, this relationship is decidedly more pronounced during the Fjord deployment (correlation of 0.95) than during the Snow (-0.44). This difference potentially arises from the fact that measurement operations at the Fjord site occasionally involved sliding the ISE-CUBE stack, during which time the incoming ventilation tubing for the Analyzer module would often come disconnected. This incoming tubing was connected to the exhaust of the Pump module, in an effort to pre-condition (i.e. slightly warm) the ventilating air. Upon disconnection,





**Table 2.** Analyzer temperature statistics in °C. 25$^{th}$, 50$^{th}$ (median), and 75$^{th}$ percentiles of DAS ($T_{DAS}$), Cavity ($T_C$), and Warm Box temperatures ($T_{WB}$) from laboratory and field. Interquartile range (IQR) provided for the same.

| (°C) | | Percentile | | | |
| | | 25$^{th}$ | 50$^{th}$ | 75$^{th}$ | IQR |
|---|---|---|---|---|---|
| $T_{DAS}$ | Lab | 46.6 | 47.5 | 49.0 | 2.4 |
| | Field | 30.8 | 35.6 | 42.1 | 11.3 |
| $T_C$ | Lab | 79.9986 | 80.0000 | 80.0015 | 0.0029 |
| | Field | 79.9986 | 80.0001 | 80.0015 | 0.0029 |
| $T_{WB}$ | Lab | 44.9997 | 45.0000 | 45.0002 | 0.0005 |
| | Field | 44.9993 | 45.0001 | 45.0009 | 0.0016 |

the Analyzer module would be directly exposed to cold ambient temperatures, resulting in lower analyzer temperatures. This
highlights the potential value of such a pre-conditioning of the ventilation air for stable analyzer temperatures.

A second environmental factor that may have a connection to analyzer temperature and temperature stability is the wind
speed. Wind speeds were relatively low at the Snow site (2 to $6\,\mathrm{m\,s^{-1}}$), but ranged up to $13\,\mathrm{m\,s^{-1}}$ during the Fjord deployment
(Figure 6b). Daily DAS IQR (Figure 6b, green dashed line) and mean daily wind speed (brown line) shows the largest IQR
values occur during days with low wind speeds. When considering all days, the two curves have a moderate anti-correlation
($-0.66$), indicating higher $T_{DAS}$ variability on days with low wind speeds. It is likely that this is an indirect relationship,
influenced more by the impact that wind speed had on the extent of operations that we did on any particular day. On days with
higher winds, we kept operations to a minimum, simply performing routine checks and switching collection vials in the Cold
Trap, both of which involved removing the protective canvas cover for a only short time (see Figure C1). However, during
less windy days (Figure C2), our operations were more extensive. We would conduct multiple profiles with the cover removed
and the analyzer temperature variability would increase. This result points to the system's potential value as a more fixed
measurement platform, one without daily disturbances. It also highlights the benefit of having a protective cover to serve as a
windbreaker for the stack modules. Therefore, after considering this wind speed/operation relationship, the direct impact that
the wind speed has on analyzer temperature is negligible in comparison.

In summary, the ambient conditions (especially temperature) did generally affect the internal analyzer temperature, and
lead to more variable and overall colder $T_{DAS}$. However, the $T_{DAS}$ remained within its nominal range of operation, with the
analyzer remaining functional for the entirety of the two deployments. We now continue our investigation with the conditions
in the measurement cavity, the most critical element of the analyzer.

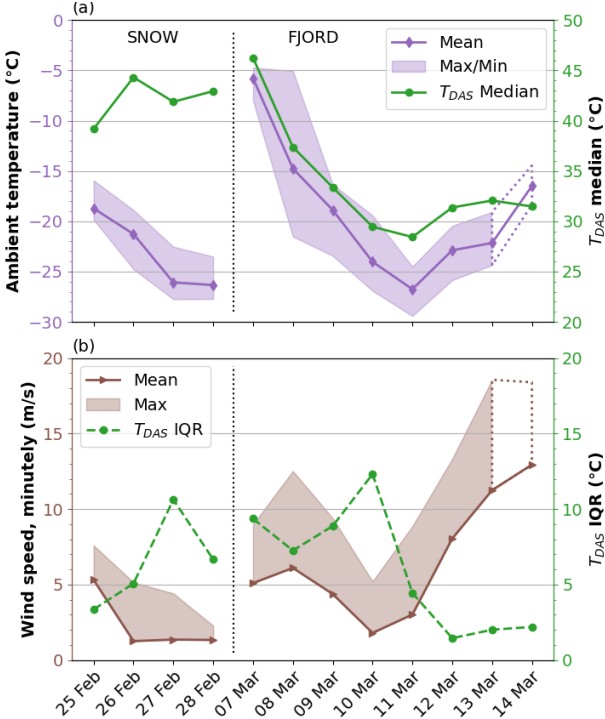

**Figure 6.** Analyzer response to ambient conditions from the two measurements sites of the ISLAS2020 field campaign (Feb: Snow, Mar: Fjord). a) Ambient daily temperatures and daily median $T_{DAS}$, as well as b) daily wind conditions alongside $T_{DAS}$ IQR. Median $T_{DAS}$ (green line) show a weak correlation (0.36) with mean ambient temperature at the deployment site (purple line), while interquartile range (IQR) of $T_{DAS}$ (green dashed line) exhibits a stronger anti-correlation (-0.66) with mean daily wind speed (brown line). Wind and ambient temperature measurements on 14 Mar affected by sea spray/icing, therefore daily values shown are taken to be representative, but actual daily values may differ.

### 4.1.2 Cavity temperature and pressure

The precision and accuracy of the temperature inside the measurement cavity of the analyzer is of the utmost importance

for precise spectroscopic measurements. For this reason, the analyzer regulates the cavity temperature very precisely about $80.00 \pm 0.01\,°\mathrm{C}$ (Steig et al., 2014). Median $T_C$ for the entire field deployment was $80.0001\,°\mathrm{C}$ (Table 2), only $0.0001\,°\mathrm{C}$ higher than the target and our laboratory benchmark value, and indistinguishable from the laboratory (Figure 5b). Additionally, IQR from field and lab were identical to $0.1\,\mathrm{mK}$ (Table 2). Across the aforementioned $T_{DAS}$ bins, $96\,\%$ of the variation about the $80\,°\mathrm{C}$ target was within $0.005\,°\mathrm{C}$ while in the lab. This is the same extent of variation that is present in the field data

(whiskers of Figure 7a, $98^{\mathrm{th}}$ and $2^{\mathrm{nd}}$ percentiles). The median of the cavity temperature from the field (black ticks) is below target at the extreme ends of DAS operating conditions ($-0.0005\,°\mathrm{C}$ in the 15 to $20\,°\mathrm{C}$ range and $-0.0002\,°\mathrm{C}$ in the 50 to $55\,°\mathrm{C}$ range). However, these bins together only constitute $4.4\,\%$ of the total observations, and are still within acceptable limits. Furthermore, IQR from the field (Figure 7a, red boxes), including the two extreme bins, is of the same magnitude ($\pm 0.0015\,°\mathrm{C}$)





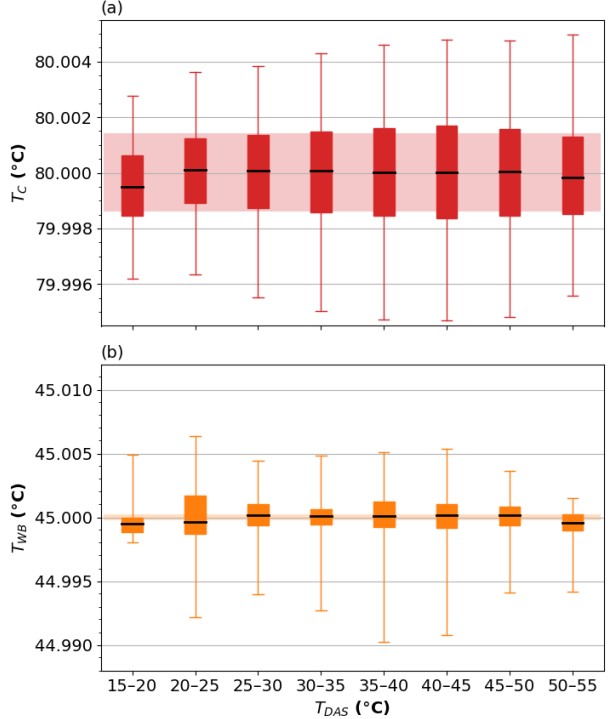

**Figure 7.** (a) CRDS analyzer cavity temperature as a function of DAS temperature. Cavity temperature target is $80.00 \pm 0.01\,°C$. Daily median indicated with black tick inside solid colored bar, upper and lower extents indicate 75th and 25th percentiles, respectively. Whiskers denote 98th and 2nd percentiles. Background shading denotes IQR from laboratory benchmark. (b) Same as (a), but for the Warm Box temperature with a target of $45\,°C$.

as the IQR of the laboratory (shaded area). Therefore, $95.6\,\%$ of field observations are made with cavity temperatures typical

of the laboratory, with the remaining $4.4\,\%$ being only slightly atypical in the median, but still within specified limits.

The pressure inside the analyzer cavity must be maintained at $66.66 \pm 0.10\,hPa$ (Steig et al., 2014). This range is maintained for $99.95\,\%$ of the field deployment (Figure 5c). Since the ISE-CUBE system does very little to modify the native flow pattern of the analyzer, we expect that the $p_C$ exhibits no dependence on the DAS temperature, nor the field deployment in general. Indeed, on any of the 12 deployment days, the $p_C$ was within the $0.1\,hPa$ specification $99\,\%$ of the time, with a maximum IQR

of $0.048\,hPa$. Across the same DAS bins used in cavity temperature analysis, the maximum departure of $p_C$ median from the target is $0.0005\,hPa$. However, when the entire deployment as a whole is considered, the distribution about the target is broader in the field as compared to the lab (Figure 5c). In fact, variability about the target (IQR: $0.035\,hPa$) is $75\,\%$ greater than what is typically found during the laboratory benchmark (IQR: $0.020\,hPa$).

After further investigation into other potential differences between the laboratory and field setups, it was determined that

this difference in cavity pressure stability stems from the enhanced sample flow configuration of the analyzer while being used





in the field (Sect. 2.2). The fast-response configuration has been used in a previous field deployment (Chazette et al., 2021), and is within the scope of the standard operating procedures.

In summary, the accuracy and precision of cavity pressure and temperature remained within specified limits. In particular, the cavity temperatures were nearly indistinguishable between laboratory and field. We therefore expect reliable functioning
of the measurement cavity of the analyzer during the field deployment. As a final analyzer parameter, we now investigate the warm box temperature.

### 4.1.3 Warm Box temperature

The WLM is part of the analyzer's laser control loop and is continuously used to target the desired wavelengths, reducing instrument drift (Crosson, 2008; Gupta et al., 2009). It is contained within the Warm Box, which the analyzer regulates the
interior temperature of to $45\,°C$. In our laboratory benchmark, more than $50\,\%$ of the variation about this target was within $0.0003\,°C$ (Table 2). Considering the entire field deployment, $T_{WB}$ distribution is broader in the field as compared to the laboratory (Figure 5d), with an IQR approximately three times larger (Table 2). Considering $T_{WB}$ across DAS bins, the size of the IQR in the field was up to 6 times larger than in the laboratory (Figure 7b, orange rectangles and shaded area). Additionally, the field medians of the two lowermost and the uppermost $T_{DAS}$ (Figure 7b, black ticks) are lower than even the
25[th] percentile of the laboratory period. This indicates that the temperature inside the Warm Box is coupled to the changing analyzer temperature. Since the analyzer has a gas inlet at the back to the WLM, ambient air temperature could also more directly impact the $T_{WB}$ than other components. As an example, sudden dips and spikes in $T_{WB}$ correspond with the onset of drops and rises in $T_{DAS}$ (Figures C1 & C2). As this range of variations could potentially have an impact on measurement quality, we now assess WLM performance in more detail during lab and field.
For the evaluation of the WLM performance, we use three spectroscopy metrics that quantify the difference between an expected model spectrum versus the fitted absorption spectrum that is actually measured by the analyzer (Johnson and Rella, 2017). The baseline shift (BS) describes the absolute value change of the spectral baseline, the slope shift (SS) indicates the change in the slope of the baseline (Johnson and Rella, 2017; Weng et al., 2020), and the residual (RS) represent the residual errors present in the fit spectrum, compared to the expected spectrum. The spectra have a first-order dependency on mixing
ratio with resulting baseline differences (Aemisegger et al., 2012; Steen-Larsen et al., 2013, 2014b; Bonne et al., 2014; Weng et al., 2020). To account for this dependency on the isotopic concentrations derived from the spectra, data from both laboratory and field have been sorted into $300\,ppmv$ bins, from 300 to $3900\,ppmv$.

Figure 8 displays a comparison between the three metrics during laboratory against field operation. Ideal analyzer performance would produce a strong correlation in the three metrics between laboratory and field data, while also aligning closely to
a 1-to-1 line. The three metrics have correlations of 0.994 (BS), 0.980 (SS), and 0.959 (RS). The BS and SS also have linear regression line slopes near to 1 ($0.90 \pm 0.06$ and $1.04 \pm 0.14$, respectively, Figure 8a and b). The RS error has the linear regression with the largest deviation from 1, ($0.75 \pm 0.15$) with an intercept of $0.05 \pm 0.03$ (Figure 8c). While analyzer performance to first order appears as desired, the RS error of the lowest humidities indicates that the spectral fit in the field deviated further compared to the lab. Thus, measurements at low humidity could be less accurate measurements in the field.





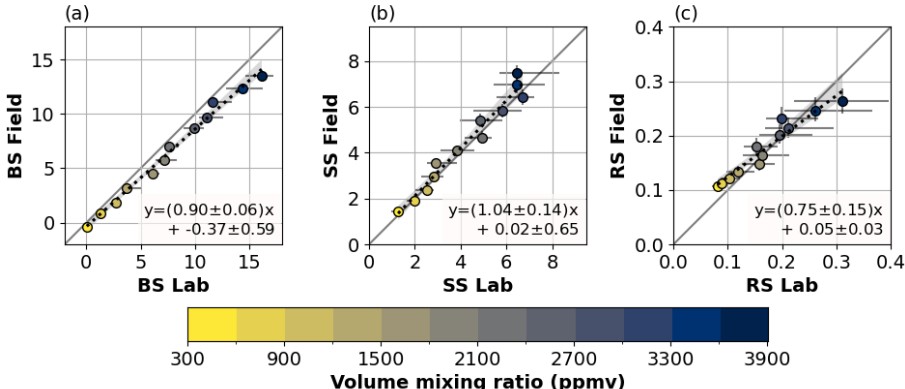

**Figure 8.** (a) Baseline Shift, (b) Slope Shift, and (c) Residuals of the expected model spectrum versus the fitted absorption spectrum from the analyzer's WLM. Light gray line represents 1-to-1 ratio. Data are divided into 300 ppmv bins, from 300 to 3900 ppmv (colorbar), with error bars denoting IQR for both lab and field measurements. Dotted lines indicate linear regressions through the data points, with gray shading showing the 95 % confidence interval.

Interestingly, the opposite is suggested at the higher humidities ($>3300$ ppmv), with a tendency for higher RS values in the lab (i.e. better spectroscopic fitting in the field). Such a peculiar result can be reconciled by considering what other factors can influence the spectroscopy. Aside from the mixing ratio, the spectroscopy can also be affected by the exact composition of the air or carrier gas, including the presence of organic contaminants (Johnson and Rella, 2017). In our laboratory benchmark, both pure nitrogen ($100\%$ $N_2$) and synthetic air ($80\%$ $N_2$, $20\%$ $O_2$) were used as carrier gases, which can cause spectroscopic
differences from our field deployment. Consequently, a precise match is unlikely. Therefore, unlike analyzer metrics such as temperatures and pressure, deviations of WLM metrics from the expected 1-to-1 relationship do not allow a final conclusion with regard to the effect of WLM temperature variation on field measurements.

     As the difference between our laboratory benchmark and field deployments prevent further conclusions, we now use measurements with our analyzer during a short-term deployment at the Zeppelin mountain observatory (475 masl), 2 km SSW
of Ny-Ålesund. This intermediate operation in a semi-regulated environment provides us with a secondary benchmark in the WLM performance evaluation. The analyzer was installed at Zeppelin observatory from 29 Feb 2020 to 3 Mar 2020, without the ISE-CUBE system and measuring ambient air. When similarly comparing the WLM metrics from the field deployment with those of the observatory, the results and uncertainties are closer to expectations (Table 3 and Figure D1). Correlations of all spectroscopic metrics are also improved. Additionally, the distribution of $T_{WB}$ during observatory (Figure 9a, thick gray)
and field (orange) deployments are similar. However, $T_{DAS}$ in the observatory (Figure 9b) are much more similar to laboratory conditions (dashed), than to field (green). This indicates that $T_{WB}$ variability is not only controlled by ambient analyzer temperatures. Finally, $T_C$ distributions are similar across all three locations (Figure 9c), demonstrating that the cavity was able to maintain temperature within an acceptable range, regardless of deployment location.





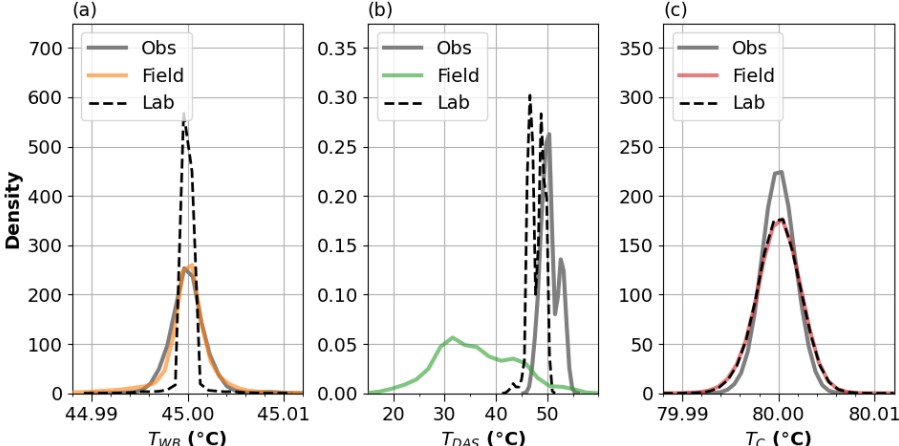

**Figure 9.** Kernel density estimations for (a) Warm Box temperature, (b) DAS temperature, and (c) Cavity temperature. Colored, thick, and dashed lines represent values from field, observatory, and laboratory, respectively. Curves have been smoothed to improve visualization. Note that "Field" curves overlap closely with with "Obs" (Observatory) and "Lab" (Laboratory) curves in (a) and (c), respectively.

**Table 3.** Correlations and linear regression values (slope and intercept) for WLM metrics (Baseline shift, BS; Slope shift, SS; and Residuals, RS) in field vs. laboratory and observatory.

|  |  | Field vs. | |
|---|---|---|---|
|  |  | Laboratory | Observatory |
| **BS** | Correlation | 0.994 | 0.999 |
|  | Slope | $0.90 \pm 0.06$ | $1.02 \pm 0.03$ |
|  | Intercept | $-0.37 \pm 0.59$ | $0.07 \pm 0.21$ |
| **SS** | Correlation | 0.980 | 0.999 |
|  | Slope | $1.04 \pm 0.14$ | $0.84 \pm 0.02$ |
|  | Intercept | $0.02 \pm 0.65$ | $0.24 \pm 0.13$ |
| **RS** | Correlation | 0.959 | 0.990 |
|  | Slope | $0.75 \pm 0.15$ | $0.96 \pm 0.09$ |
|  | Intercept | $0.05 \pm 0.03$ | $0.03 \pm 0.02$ |

Then to summarize, the measurement cavity remained mostly unaffected, and the warm box maintained a temperature at the
same quality as of a field laboratory. We can now therefore proceed to quantify the quality of the water isotope measurements
from the ISE-CUBE system.



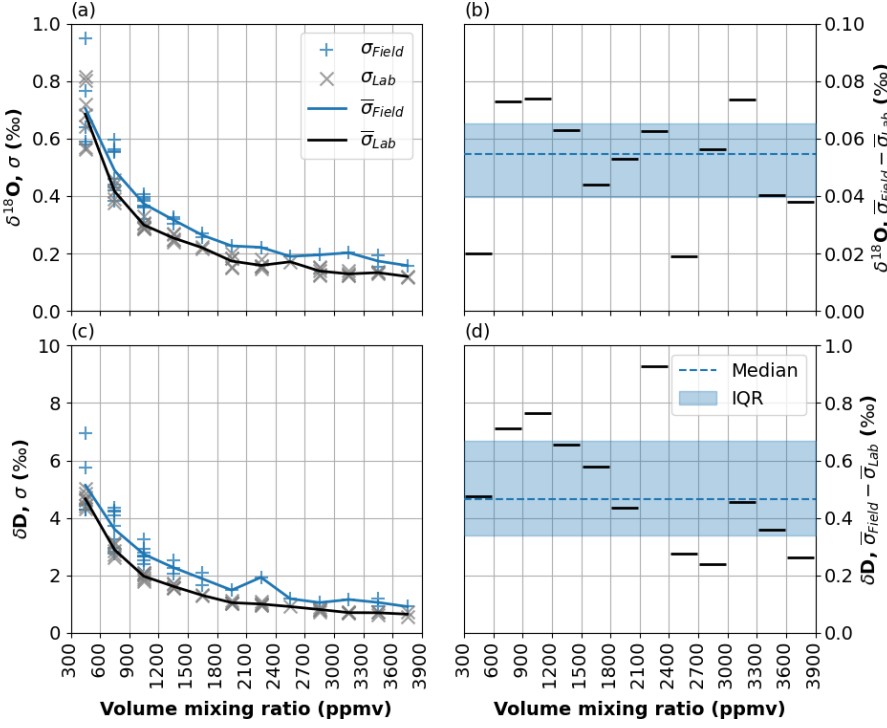

**Figure 10.** Standard deviations of $\delta^{18}$O and $\delta$D in laboratory and field, alongside differences between the two, across $300\,\mathrm{ppmv}$ bins, from $300$ to $3900\,\mathrm{ppmv}$. (a) Standard deviations of $\delta^{18}$O over $5\,\mathrm{min}$ periods from field (blue '+') and laboratory (gray 'x'); see text for details. Bin means depicted with blue (field) and black (laboratory) lines. (b) Difference between bins means from field and laboratory. Dashed blue line is the median, while shading shows IQR. (c,d) Same as (a,b) but for $\delta$D.

## 4.2   Measurement quality of water vapor isotopes

Based on the assessment of analyzer parameters presented above, measurement conditions within the ISE-CUBE system and in the laboratory differ mostly with respect to the variability of $T_{WB}$ and $T_{DAS}$. To identify a potential impact of this temperature variability on the vapor isotope measurements, we now compare the variability of the isotopic signal between laboratory and field. Since mixing ratio (and thus the amount of molecules in the measurement cavity) is a key factor in the precision of the CRDS measurements, we divide the measurement data into the same bins as we used before (Section 4.1.3). For each bin and for each day in the field, we identify the $5\,\mathrm{minute}$ period with the most stable mixing ratio (i.e. lowest standard deviation); the mixing ratio must stay within the bin range for at least $50\,\%$ of the time during this $5\,\mathrm{minute}$ window (i.e. necessary for periods close to bin edges). Within the laboratory benchmark, rather than daily periods, we focus on eight distinct usage events conducted for instrument characterization purposes. During these usage events, the analyzer was subjected to step-wise mixing ratio sequences between $400$ and $3900\,\mathrm{ppmv}$ (FigureE1). The steps usually lasted $5$ to $10\,\mathrm{minutes}$, and the sequences did not necessary follow a consistent step magnitude. These sequences used a standard of a similar depletion ($\delta^{18}$O: $-40.02\,‰$,







$\delta$D: $-307.79\,‰$) as our field measurements. By identifying periods this way, we prevent over-sampling of low mixing ratios

or specific field days, while still generating a sample population representative of mixing ratios encountered in the field. The 1-$\sigma$ 5 minute standard deviation over each of these periods was then calculated for both $\delta^{18}$O and $\delta$D.

For the laboratory and field periods, and for both $\delta^{18}$O and $\delta$D, the measurement precision decreases with decreasing mixing ratio (Figure 10a,c). While standard deviations of the laboratory are approximately constant above $3000\,\mathrm{ppmv}$, in the lowermost bin of $300$ to $600\,\mathrm{ppmv}$, 5-min standard deviations in the field reach up to $0.9\,‰$ for $\delta^{18}$O (Figure 10a), and

$7\,‰$ for $\delta$D (Figure 10c). Field bin means (blue line) are consistently higher than laboratory bin means (black line) across all humidities. However, when considering the bin means, the difference between the laboratory and the field is never more than $0.08\,‰$ (Figure 10b) and $1.0\,‰$ (Figure 10d), for $\delta^{18}$O and $\delta$D, respectively. The overall median difference across all bins is $0.06\,‰$ and $0.47\,‰$ for $\delta^{18}$O and $\delta$D.

In summary, the two field deployment locations exhibit consistently lower measurement precision of both isotope species.

This lower precision in the field is likely a consequence of the more variable $T_{WB}$ and $T_{DAS}$, as well as more variable composition of the ambient air. Nonetheless, differences to measurements in a laboratory environment are quite limited, and do not hinder useful measurements, in particular if the measurement precision is quantified.

## 4.3   Cold Trap module performance

The Cold Trap expansion module was fully integrated into the sample air flow of the ISE-CUBE system during the field tests.

Since the Cold Trap box was not thermally regulated, the cryocooler unit was operated well below the manufacturers minimum operating guideline of $5\,°\mathrm{C}$, reaching $-5\,°\mathrm{C}$ over sustained periods, and even down to $-20\,°\mathrm{C}$ occasionally. Nonetheless, the cryocooler reliably provided temperatures of $-80\pm5\,°\mathrm{C}$ at the base of the vial enclosure for $80\,\%$ of the deployment. The remaining $20\,\%$ mostly occurred during overnight collections, whereby excessive frost buildup on the exterior of the brass vial enclosure would provide additional insulation and the vial temperature would drop below $-85\,°\mathrm{C}$. Regardless, water

vapor was successfully collected during a total of $28$ periods, with up to $0.3\,\mathrm{mL}$ in a single sample. Vapor collection in a vial lasted between $8\,\mathrm{hours}$ and $16\,\mathrm{hours}$ during daytime and nighttime, with target flow rates of $0.5\,\mathrm{L\,min^{-1}}$ and $0.25\,\mathrm{L\,min^{-1}}$, respectively. However, substantial ice crystal formation in the neck of the collection vials inhibited and decreased flow during multiple collection periods. This ice formation also compromised sample recovery during vial exchange, causing frozen sample to fall out of the vial during collection. These deficiencies could be corrected with a different collection vial, although sample

analysis procedure would need to be changed.

In summary, we provide here a first proof-of-concept that a cryocooler-based Cold Trap module can be integrated into the ISE-CUBE system. Remaining shortcomings of the design can be rectified in future module iterations. A more detailed evaluation of the performance of the Cold Trap module in terms of sampling efficiency and its use for in-field calibration is however beyond the scope of this manuscript and will be detailed in a future publication.





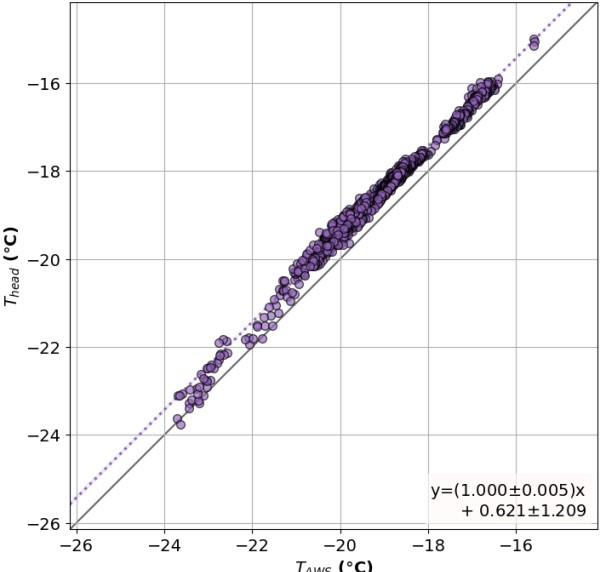

**Figure 11.** Comparison between temperature sensor of automated weather station ($T_{AWS}$) and the inlet temperature sensor ($T_{head}$). Both were at a height of $100\,\mathrm{cm}$ above the surface of the snow for $37\,\mathrm{hours}$. Solid gray line represents 1-to-1 ratio. Dotted line indicates linear regressions through the data points, with purple shading (barely visible under dotted line) shows the $95\,\%$ confidence interval.

## 4.4 Profiling module performance

The Profiling expansion of the ISE-CUBE system can be evaluated in terms of the response time for mixing ratio and isotope measurements. With the heated tubing from inlet tip to the inlet of the analyzer and a diameter of $4.5\,\mathrm{mm}$ at a flow rate of about $9\,\mathrm{L\,min^{-1}}$ in the Profiling module tubing, and $0.15\,\mathrm{L\,min^{-1}}$ in the manifold tubing, the minimum response time would be approximately $6\,\mathrm{s}$. However, estimates from the field show the actual response time of the analyzer to a humidity change to be $13$ to $18\,\mathrm{sec}$. Following the approach of Steen-Larsen et al. (2014b) and Wahl et al. (2021), a fast Fourier transform analysis documented a practical minimum averaging time of approximate $30\,\mathrm{s}$ due to wall effects in the tubing and analyzer causing increased memory effect. More detailed evaluation of the response of the inlet is currently precluded by the lack of a suitable device to produce a defined vapor isotope stream at sufficiently high flow rates.

In addition, the auxiliary measurements from the ultrasonic distance sensor and the temperature probe at the tip of the inlet can be evaluated. The distance given by the ultrasonic sensor was periodically checked against a manual tape measure throughout the field deployment (accuracy $\pm\sim 2\,\mathrm{cm}$). The behavior of the ultrasonic sensor did vary between the two measurements sites, likely as a result of underlying surface. While deployed over snow, the sensor functioned with $2\,\mathrm{cm}$ accuracy over the range of 50 to $205\,\mathrm{cm}$. When the inlet head was lowered below $50\,\mathrm{cm}$, the sensor gave unreliable and clearly spurious measurements. We speculate that the particular type of ultrasonic sensor used was affected by the acoustic properties of the snowpack below this threshold. In these circumstances, manual distance was taken with a tape measure, with corresponding


markings made on the controlling steel cable (Figure 3). No issues with distance sensors were encountered while deployed at the Fjord location with water or sea ice at the surface. At both locations, a low signal to noise ratio and a jumpy distance signal was observed during strong winds ($>11\,\mathrm{m\,s^{-1}}$), possibly due to the ultrasonic pulse being advected away before the reflected signal was received.

Finally, the temperature sensor above the tip of the inlet of the profiling module was compared against the temperature sensor of an automated weather station (Jocher et al., 2012; Schulz, 2017) ($20\,\mathrm{m}$ away) at a height of $100\,\mathrm{cm}$ for $37\,\mathrm{hours}$ (during 25 Feb and 26 Feb). The minutely averaged temperature records of the two sensors show a high correlation of 0.991, with a linear regression slope of $1.000 \pm 0.005$ (Figure 11). The profiling sensor consistently recorded higher temperatures than the AWS, having a linear regression offset of $0.6\,^{\circ}\mathrm{C}$. While distance between sensors might account for some of this discrepancy,
the gradients observed with the Profiling module are almost an order of magnitude larger, with relative changes well captured due to the high linearity. Overall, the Profiling Module functioned reliably, guiding ambient air to the analyzer with a response time for water vapor step changes within 13 to $18\,\mathrm{sec}$.

## 4.5    Example of a profiling operation

We now present an example for a profiling operation in the surface layer during strongly stable conditions, performed at heights
between $0.04\,\mathrm{m}$ and $2.00\,\mathrm{m}$ above snow on the morning of 28 Feb 2020. The entire profiling operation consisted of a sequence of 6 steps between the lowest level at $0.04\,\mathrm{m}$, progressing to $2.00\,\mathrm{m}$, with about 5 min time intervals (Figure 12a, black bars). The specific humidity during this upward profile increased from $0.25\,\mathrm{g\,kg^{-1}}$ at the lowest level, to $0.4\,\mathrm{g\,kg^{-1}}$ at $1.50\,\mathrm{m}$ and above (Figure 12a, blue dashed). The calibrated isotope species showed the strongest gradients in the lowest $1\,\mathrm{m}$ above the surface, with $-0.5\,\text{‰}/\mathrm{m}$ and $17\,\text{‰}/\mathrm{m}$ respectively.

After working through the upwards stepping of our profile, we further probed the substantial moisture gradient with first a large $1\,\mathrm{m}$ step down, from $150\,\mathrm{cm}$ to $50\,\mathrm{cm}$ (Figure 12a). As we expected, this step down resulted in a drop in specific humidity; it again rose when height was increased to $120\,\mathrm{cm}$. During further narrowing iterations, we probed near the height of the strongest moisture gradient in the surface layer. Simultaneously, we captured isotope signature of $\delta^{18}\mathrm{O}$ and $\delta\mathrm{D}$ (Figure 12b). At 9:07 there was a sudden veering of the winds (S to SW), followed by a backing (SW to ESE) (Figure 12e). The veering
was also associated with an increase in wind speed, and the development of shear in the lowermost meter (Figure 12d). In addition, the already present temperature gradient strengthened further (Figure 12e). From 9:30 until 9:55, the temperature gradient between the two heights (250 to $100\,\mathrm{cm}$ and 100 to $50\,\mathrm{cm}$) converged as observed with continuing profiling operations. Ultimately, after profile completion at 9:55, we left the inlet head at a static height of $75\,\mathrm{cm}$ for some hours (not shown), in an effort to observe any oscillations moving through the interface. Duplicating such a process with a valve/manifold system in the
surface layer would involve at least a tripling of inlet tubing, yet without guarantee that any given inlet would be at a height of interest.



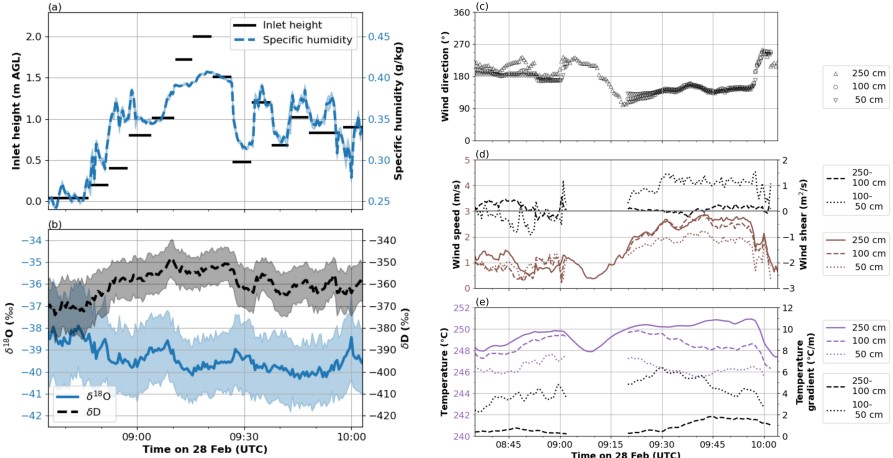

**Figure 12.** $30\,\text{s}$ averaged timeseries of profiling operations, alongside ambient wind and temperature conditions from the morning of 28 Feb 2020. (a) Inlet head height (above surface of the snow; solid black) and specific humidity (dashed blue). Shading denotes 1-$\sigma$. (b) $\delta^{18}$O (solid blue) and $\delta$D (dashed black). Shading denotes 1-$\sigma$. (c) Wind direction: $250\,\text{cm}$ = upwards triangles; $100\,\text{cm}$ = circles; and $50\,\text{cm}$ = downwards triangles. (d) Wind speed (left axis, colored): $250\,\text{cm}$ = solid colored lines; $100\,\text{cm}$ = dashed colored lines; and $50\,\text{cm}$ = dotted colored lines. Wind shear (right axis, black): $250-100\,\text{cm}$ = black dashed lines; and $100-50\,\text{cm}$ = black dotted lines. (e) Temperature (left axis, colored): $250\,\text{cm}$ = solid colored lines; $100\,\text{cm}$ = dashed colored lines; and $50\,\text{cm}$ = dotted colored lines. Temperature gradient (right axis, black): $250-100\,\text{cm}$ = black dashed lines; and $100-50\,\text{cm}$ = black dotted lines.

## 5 Conclusions and Outlook

In this work, we have detailed the design and performance of the new ISE-CUBE system for in-situ vapor isotope measurements. The ISE-CUBE system is a modular field-going deployment system for in-situ laser-based spectroscopy. The modular
design enables rapid installation, while the compact system size provides minimized flow distortion around the measurement site. We apply the system and evaluate its performance for the specific case of stable isotope measurements of water vapor in the lowermost $2\,\text{m}$ of the atmosphere in an Arctic environment.

During a two-week long field deployment at Ny-Ålesund, Svalbard, Norway during Feb–Mar 2020, the analyzer encountered extreme ambient winds and temperatures. While ambient temperatures reached down to $-30\,^{\circ}\text{C}$, and winds were gusting to
over $20\,\text{m s}^{-1}$, the analyzer remained within its specified range of measurement conditions with regard to $T_C$, $T_{WB}$, $T_{DAS}$, and $p_C$. Measurement precisions during the field deployment were on average $0.06\,‰$ ($\delta^{18}$O) and $0.47\,‰$ ($\delta$D) lower than in the laboratory benchmark, irrespective of ambient mixing ratios. Our results demonstrate that the ISE-CUBE system allows for in-situ CRDS measurements in harsh environments that are only marginally less precise than measurements in a laboratory environment.

The profiling expansion module, a height-adjustable sampling arm within the range of $0$ to $2\,\text{m}$ above the surface, enabled flexible and sufficiently precise profiling of the water isotope composition throughout the surface layer. With a response time





of about 13 to 18 sec, giving a minimum averaging time of 30 s, the ISE-CUBE system captured profiles and gradients in the strongly stratified stable boundary layer. In addition, the flexibility of the measurement height was a clear asset for measurements over water surface with strong tides. Additionally, combinations with tower and manifold systems are also possible, thus

supplementing high-resolution vertical profiles in the surface layer with sampling across a larger height range.

Integration of the Cold-Trapping expansion module based on the design of Peters and Yakir into the ISE-CUBE system enabled quantitative vapor collection. Samples were typically collected over a duration of 8 hours or more in low-humidity environments, resulting in a maximum collection of 0.3 mL at a time. Such cold-trapping enables subsequent liquid sample analysis in a laboratory environment for quality control of the calibration, and the measurement of $H_2^{17}O$ for triple-isotope

capability. While the cold trap module is functional, additional evaluation of the sampling efficiency, and design optimizations to facilitate sample handling are needed.

The next expansion module would focus on an in-field calibration system. In general, there are multiple potential calibration devices using a variety of vapor generation methods (Iannone et al., 2009; Gkinis et al., 2010; Ellehoj et al., 2013); the device just needs to be robust and suitably compact, all while generating a consistent source of known vapor. For example, Leroy-Dos

Santos et al. (2021) have put forward an instrument that can generate stable vapor streams down to 70 ppmv, one of which has been operating mostly autonomously in Antarctica, with little manual intervention. Ultimately, when integrating a future calibration module into the system, the most important considerations are for the research aims and logistical limitations of the deployment itself.

One current limitation of the ISE-CUBE system is the lack of an active temperature control capability, which presently

prevents operation in warmer measurement environments. With the modular approach, the addition of a dedicated ventilation module with larger fans and active cooling/heating unit is straightforward, and would enable deployment in warmer and more variable climatic conditions. An additional battery power module would permit mobile operation for limited time periods, thus allowing for the study of spatial variability.

The availability of a modular, versatile deployment system such as the ISE-CUBEs implies easy access to remote locations

and environments, while maintaining necessary data quality standards. Locations may include, but are not limited to, glaciers, sea ice, lakes, coastal areas, caves, forests, grasslands, croplands, deserts, and other places where evaporation and condensation interactions with the surface contribute to the vapor isotope composition of the near-surface atmosphere. The limited footprint and robust handling also allows for installation on open moving platforms to obtain spatial transects. Finally, laser spectrometers for other atmospheric trace gases, such as methane, carbon dioxide, or carbon isotopes may be integrated into the system.

As we provide the design in a easily reproducible way to the community (see Supplemental Material), we endeavor to enable further development and widespread acquisition of high-quality datasets from previously inaccessible measurement locations.



## Appendix A: ISE-CUBE component list

**Table A1.** Components used in the ISE-CUBE system. General pneumatic connections (PTFE tubing, unions, reducers, etc.) made with parts from Swagelok Inc. Files for 3D printed ventilation mounts were custom-made and not listed (see Supplemental Material). Modular abbreviations: AN=Analyzer module; PU=Pump module; CT=Cold Trap module; PR=Profiling module.

| Module | Component | Manufacturer | Model/Series | Size(LxWxH); Weight; Detail |
|---|---|---|---|---|
| AN/PU/CT | Module container | Pelican Products Inc. | iM2875 Storm | 632 x 602 x 333 mm; 9.1 kg |
| AN | CRDS analyser | Picarro Inc. | L-2130i | 20.4 kg; 230 VAC |
| AN/PU | Reinforced PVC vacuum tubing | Kuriyama of America Inc. | K7160-06 | 3/8 inch |
| PU | CRDS vacuum pump | KNF DAC GmbH | N920AP.29.18 | 10.5 kg; 230 VAC |
| PU | Inlet vacuum pump | KNF DAC GmbH | N022AN.18 | 4.0 kg; 230 VAC |
| PU | UPS | Eaton | EL500FR | 2.9 kg; 230 VAC |
| CT | Cryocooler | Sunpower Inc. | Cryotel MT | 2.1 kg; 24 VDC |
| CT | Collection vial | ThermoSci | 2-SVW Chromacol | 2 mL |
| CT | UPS | Phoenix Contact | 2866611 | 5.6 kg; 24 VDC |
| CT | 12 VDC (out) power supply | Mean Well | DDR-15G-12 | 9 to 36 VDC input |
| CT | Flow meter | Sierra Instruments Inc. | TopTrak 822 | 12 VDC; 0-1 & 0-30 $\mathrm{L\,min^{-1}}$ |
| AN/PU/CT | Ventilation fan | ebm-papst GmbH & Co. KG | RL 90 | 0.7 kg; 230 VAC |
| AN/PU/CT | Protective cover | IKEA | TOSTERÖ | 1000 x 700 x 900 mm; 0.75 kg |
| AN/PU/CT | Power connector | Amphenol | 62GB | 230 VAC |
| AN | Data connector | Amphenol | RJ45F7RJ | RJ45 |
| AN/CT | Data connector | RS Pro | 111-6759 | USB |
| AN/PU/CT | Inlet vacuum connector | Swagelok Inc. | SS-400-61 | 1/4 inch |
| AN/PU | Picarro vacuum connector | Swagelok Inc. | SS-600-61 | 3/8 inch |
| AN/PU | One-way check valve | Swagelok Inc. | 6L-CW4S4 | 1/4 inch |
| PU/CT | Aluminium support frame | RatRig | V-Slot 2020 | |
| CT/PR | Microcontroller | Arduino | UNO & Mega | 12 VDC |
| CT/PR | Temperature sensor | Velleman | VMA324 | $-55$ to $125\,^\circ$C |
| PR | Ultrasonic distance sensor | SparkFun Elec. | SEN-15569 (HC-SR04) | 0 to 5 m |
| PR | Module Container | Pelican Products Inc. | 1120 | 214 x 172 x 98 mm; 0.6 kg |
| PR | Inlet heat trace | Thermon Inc. | BSX 10-2 | 60 $^\circ$C; 32 W/m |
| PR | Aluminium tripod | Campbell Scientific | CM110 | 15 kg |
| PR | Hand winch | Hamron | - | 350 kg rating |
| AN/PU/CT | AC mains power cable | Lapp | 0013631 | 3-core; 230 V; $-40\,^\circ$C rating |
| PU/PR | DC power & signal cable | Alpha Wire EcoFlex | 79002 SL005 | 3-core; 12 V; $-40\,^\circ$C rating |





# Appendix B: Campaign location

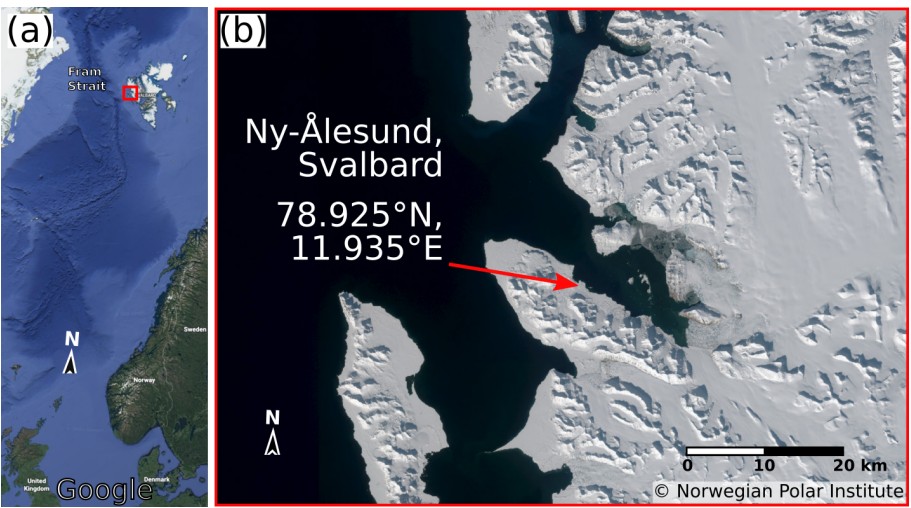

**Figure B1.** (a) Location of Svalbard and campaign area (red square) (© Google Earth 2022). (b) Zoom in on campaign area showing the location of Ny-Ålesund on the southern shore of Kongsfjorden (© Norwegian Polar Institute, http://toposvalbard.npolar.no).





## Appendix C: $T_{DAS}$ and $T_{WB}$ relationship

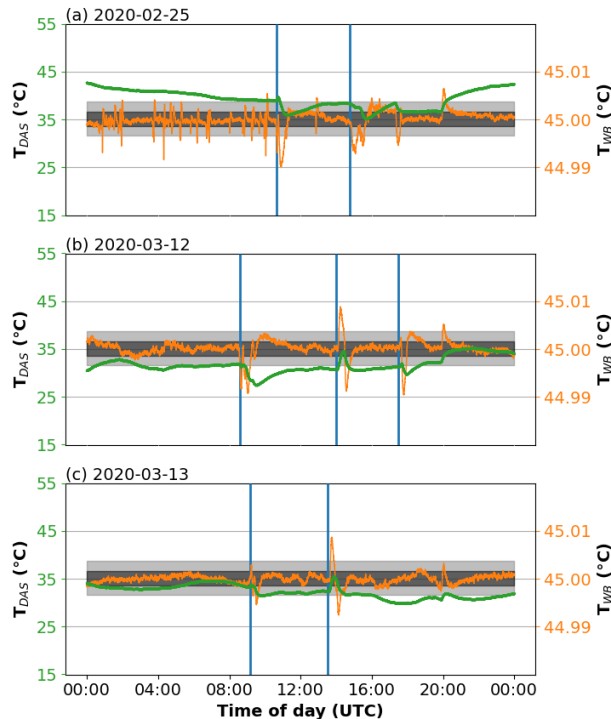

**Figure C1.** Daily timeseries of DAS temperature (left axis, green) and Warm Box temperature (right axis, orange) on (a) 25 Feb 2020, (b) 12 Mar 2020, and (c) 13 Mar 2020. Black shading denotes spread between 2nd and 98th percentiles of Warm Box temperatures during laboratory benchmark. Gray shading indicates the same for the time in the Zeppelin Observatory.

515  Increases or decreases in $T_{DAS}$ had a tendency to initiate sudden dips or spikes in $T_{WB}$ (Figures C1 & C2). While the impact of $T_{DAS}$ perturbations could last up to around 4 hours, $T_{WB}$ instability was shorter lived, usually returning to benchmark values within an hour. However, the magnitude of these perturbations, and the subsequent recovery time, is largely related to the kind of operation prompting the disturbance. Brief operations (Figure C1), such as routine checks or swapping Cold Trap collection vials, would produce a relatively small change in $T_{DAS}$, as compared to longer operations (Figure C2), such as

520 profiling. Accordingly, recovery time for the $T_{WB}$ would also be longer during the profiling periods, though the magnitude of the $T_{WB}$ dip/spike could be comparable across all operations.

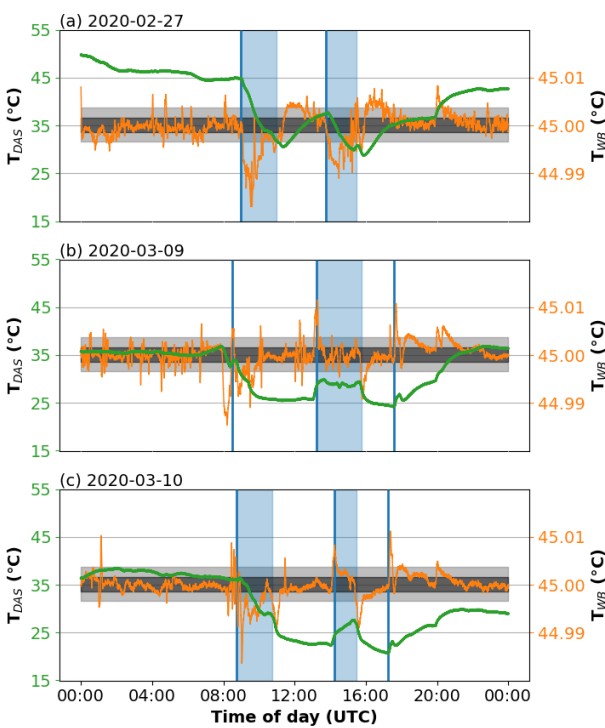

**Figure C2.** Similar to Figure C1, but for (a) 27 Feb 2020, (b) 9 Mar 2020, and (c) 10 Mar 2020. Blue shading indicates profiling periods, where the ISE-CUBEs were attended with the cover removed.



## Appendix D: WLM metrics between field and observatory

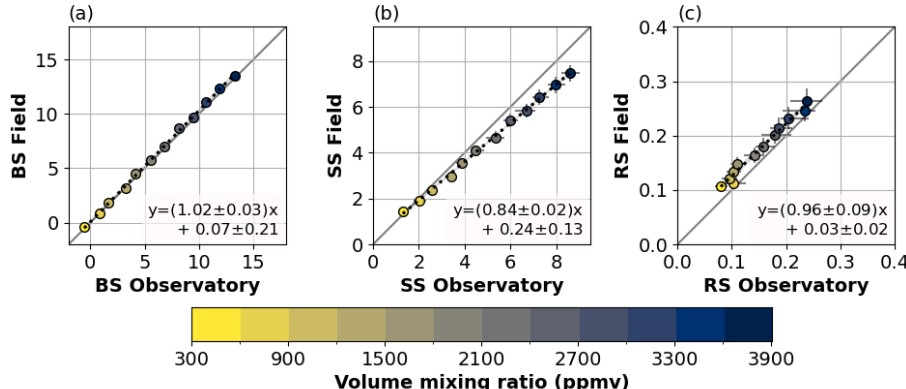

**Figure D1.** (a) Baseline Shift, (b) Slope Shift, and (c) Residuals of the expected model spectrum versus the fitted absorption spectrum from the analyzer's WLM. Light gray line represents 1-to-1 ratio. Data are divided into $300\,\mathrm{ppmv}$ bins, from $300$ to $3900\,\mathrm{ppmv}$ (colorbar), with error bars denoting IQR for both Zeppelin observatory and field measurements. Dotted lines indicate linear regressions through the data points, with gray shading showing the $95\,\%$ confidence interval.





## Appendix E: Laboratory benchmark sequences

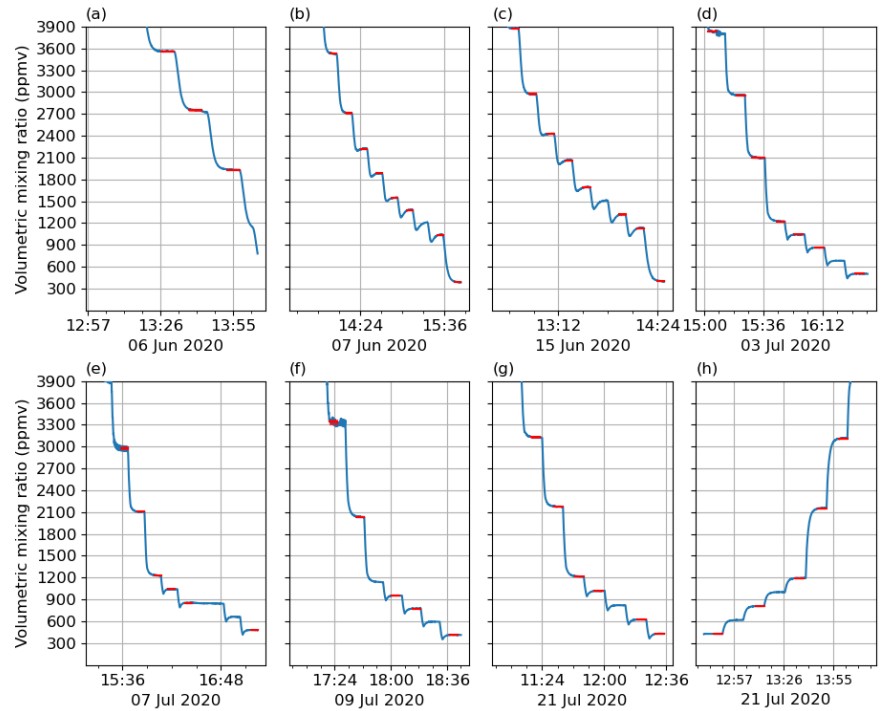

**Figure E1.** Humidity step sequences from eight (a-h) distinct usage events from laboratory characterization of the analyzer, during the period of June to July 2020. Blue line depicts volumetric mixing ratio, with red highlights showing the most stable 5 minute periods within each mixing ratio bin.



*Author contributions.* Conceptualization (design): AS, HS, HCSL; Methodology (construction): AS, HS; Investigation: AS, HS; Formal
analysis: AS, HS; Writing – original draft preparation: AS; Writing – review and editing: All authors

*Competing interests.* The authors declare no conflict of interest. The funders had no role in the design of the study; in the collection, analyses, or interpretation of data; in the writing of the manuscript, or in the decision to publish the results.

*Acknowledgements.* The authors wish to acknowledge: Anak Bhandari, Helge Bryhni, & Tor de Lange for their technical expertise during the construction process; Alexander Schulz for the data from the AWI Ny-Ålesund eddy-covariance meteorological station; and Aina Jo-
hannessen, Alena Dekhtyareva, & Marius Jonassen for their roles in the ISLAS2020 campaign. The authors also wish to make a special acknowledgment to the memory of Leon Isaac Peters for his incredibly patient and helpful insights regarding the Cold Trap Module.





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
