# Peer review of "A modular field system for near-surface, vertical profiling of the atmospheric composition in harsh environments using cavity ring-down spectroscopy"

_Atmospheric Measurement Techniques, 2022_

## Referee Comment (RC1)

**Review of "A modular field system enabling cavity ring-down spectroscopy of in-situ vapor observations in harsh environments: The ISE-CUBE system"**
By Andrew Seidl et al. AMTD

This paper presents a new compact packing system for a commercial cavity ring-down laser spectrometer that is intended for in-situ deployments in harsh (here cold, polar) environments. The manuscript also presents a few additional modules such as a cold trap module and a profiling arm. While I find the paper well-written and very interesting to read, I have a few fundamental critiques in particular with respect to the structure and focus of the paper that the authors should reflect upon and address before acceptance of the manuscript.

1) **Technical innovation and significance:** While I really like the level of technical detail and completeness of the description of the ISE-CUBE, I do not fully understand why it stands in the center of a paper publication. Dozens of previous scientific investigations have been conducted in different in-situ installations of cavity ring-down spectrometers in containers, cars, ULM, ventilated aluminum housings, tents or aircraft racks. All these deployments were done in such a way as to address the scientific question at hand in the best possible way. The ISE-CUBE seems a useful packing method for exactly the chosen deployment: namely near-surface profiling of stable water isotope gradients in cold environments. But already in the midlatitudes and especially in the tropics the chosen setup would not work due to overheating. In my view the technically relevant and innovative part of this study is not the CUBE but the profiling arm, which however is only very sparsely addressed. Therefore, in my view the profiling arm should stand in the center of the story framing. The full use of the compact packing provided by the ISE-CUBE only becomes obvious, when combined with the profiling arm. The authors should seriously reconsider their storyline, provide a better literature-overview of existing studies with different in-situ deployments, and justify why such a detailed presentation of a very specific packing is useful to the community. To me the fact that the system is not autonomous in terms of power use is a big drawback and doesn't make the system so much more flexible than a sheltered installation with a long inlet-line.

2) **Motivation for a profiling system of the near-surface profiles within 2 m above the surface:**
   - As mentioned above I really like the profiling arm and think that this is a clear innovation and add-on to the current state of the art in the isotope literature. It also has in my view clear potential for scientifically relevant investigations. The authors should mention these in the introduction more explicitly: why is it important to measure near-surface humidity/isotope and potential other trace gas gradients up to 2 m height?
   - Normally bulk fluxes are computed using measurements over about 10 m depth near the surface, why are the authors interested in the lowest 2 m?
   - The authors should highlight more clearly in the introduction why in a polar environment it is of utmost importance to have short inlet lines (due to strong interactions with the tubing walls at low concentrations, longer response times, lower precision at low concentrations).
   - What makes a profiling arm with free choice of sampling height more valuable than a setup with a manifold and inlets at discrete heights? This is an essential argument

for the profiling arm and should come very early in the manuscript. It is now mentioned only at the very end at L. 465.

3) **Section 4.1 & 4.2:** this section is much too detailed: 7 pages to state that the measurements were essentially unaffected by the harsh environmental conditions seems exaggerated to me. I am conscious of the effort that the authors put into the data analysis to come to that conclusion (stated at L. 374-376) and I fully acknowledge that this effort is worthwhile. Figures 5 to 7 with respective tables and shortened text would make an excellent supplement. But the information given in the paper should be succinct. The DAS temperature is not that relevant for the measurements, much rather the cavity temperature and pressure should be kept stable (this can be summarized in a few sentences). The WLM discussion in Section 4.1.3 remains inconclusive to me. The importance of the air prewarming by using the exhaust of the pump module can be mentioned in the methods section. A maximum 1-page summary of these results putting forward mainly the results of Section 4.2 (L. 399-402) with Fig. 10 should be sufficient to describe the main results and keep the reader's attention focused.

4) **Cold trap module:** This is an interesting adaptation of the Peters & Yakir 2010 system. However, if no data from this system is shown and compared to the CRDS data, then this part should be left away. Currently, this part of the paper is difficult to assess without data.

5) **Field calibration expansion module:** I do not understand why a calibration module is useful for such a deployment, which needs manual handling of the system anyways. Recent studies have shown that CRDS systems operate reliably over the timescale of several days with minimal drift (without calibration), such that a calibration every few days (1-3) is entirely sufficient and can be done in shielded temperature regulated conditions. See also the statement of the authors themselves at L. 220.

6) **Profiling module performance:** as mentioned above, I think that the real innovation of this paper is this profiling arm, which also makes the need for a low-footprint and modular box clear to keep the length of the inlet line at a minimum. Unfortunately, the authors put much more effort in sections 4.1 and 4.2 than in the key sections 4.4 and 4.5.
- I recommend restructuring these sections and showing more results on this essential part. In my view L. 421 to 428 should be in the methods.
- The response time of the system & precision at the encountered concentration should inform about the ideal length of the measurement periods at a given height. This point should be discussed. Is 30 s averaging ideal also from a signal-to-noise ratio perspective? Or should it be longer?
- The temperature sensor calibration is a good thing to do, but a Supplement figure would be sufficient.
- Fig. 12 is (one of) the most interesting figures (together with the very nice technical drawings in Figs. 1-3) but it is difficult to read because it is shown as a time series (and too small with many panels). A profiling arm allows to measure profiles, so why not show profiles? When looking at Figure 12 and considering the main aim of the paper (providing a modular system that is able to measure near-surface isotope and trace gas profiles in cold environments) then I wonder: can the proposed setup resolve the vertical gradients given the uncertainty of the measurements at these low concentrations? The authors should show the vertical profiles under different

conditions including the total uncertainty of the measurements and discuss this very important question. Also, in addition to the isotope, temperature and wind information they should add the water vapour mixing ratio and dexcess.

**Detailed comments:**
I refrain from a detailed list of language and technical comments here, given my advice above for fundamental reorganization of the paper. I however chose to list a few points that need clarification in the text:
-   L. 15: which processes? Those relating to fluxes?
-   L. 17: during stable stratification -> really only then? I can imagine many situations in which the stratification is not stable and in which near-surface measurements would be very useful.
-   L. 20: "disentangling water vapor of different origin and undergoing different" processes -> do you want to disentangle the water vapor? Or the different sources of the water vapour?
-   L. 23: did Steen-Larsen et al. 2013 investigate moisture sources? Or airmass origin? Maybe choose a different reference here. Also Sodemann 2017 is a proposal that is not accessible online and not a document that I would expect to be referenced in a peer-review paper. Maybe Sodemann et al. 2017 is meant?
-   L. 26: remove "or so" (spoken language)
-   L. 30: what is the advantage of an in-situ system such as ISE-CUBE-profiling-arm over a line with a manifold? Please be more explicit. This touches upon the key innovation of this work.
-   L. 51: Wall effects -> indeed very important and how is that addressed? How long are the response times of the system? This is important for the profiling strategy (i.e. how long does the arm stay at a given elevation)
-   L. 101: stand-alone field operation -> no power (how much in total?) is needed
-   L. 169: I would say this is a typical example of an unstable situation at least over the open ocean.
-   L. 190: An overview … **is given**.
-   L. 204: if that is a central tool to this publication it should be made available online along with the data
-   Section 3.4: I have difficulty assessing if the comparison dataset from the lab is an adequate one to use. Is the amount of data (sample size) and sampling frequency comparable to what was used in the field? The description is a bit vague in this respect.
-   L. 387: the fact that the vials have to be manually changed in the cold trap module should be mentioned in the methods.
-   L. 353: I cannot follow the argument why the only slightly larger RS at low water vapour mixing ratios in the field necessarily implies less accurate measurements in the field.
-   L. 360-362: so then why such a detailed discussion of these metrics?
-   L. 416-419: I don't understand why the authors introduce the cold trap module if they don't use its data. That makes it difficult to assess if the system is fulfilling its purpose.
-   Fig. 11 should go to the supplement.

- L. 424: how were these response times estimated, to me the averaging intervals are also a key factor for optimizing the signal-to-noise ratio and obtaining the best possible precision.
- L. 449: "strongly stable", this is even an inversion
- L. 454: which isotope gradients do you mean here?
- L. 458: we captured d18O and dD (leave away "isotope signatures" of, that is a repetition)
- L. 462: what does "the temperature gradient… converged" mean here?
- L. 463: not shown -> but that would be very interesting!
- L. 463: this is very important and should be mentioned in the introduction as a motivation for the ISE-CUBE with profiling arm system.
- L. 483: "Flexibility of the measurement's height… with strong tides" interesting, but I missed that argument in the results part of the manuscript
- L. 486-491: the authors should compare the cold trap sampling to the CRDS measurements or leave it away.
- L. 492-498: As mentioned above, I do not understand why this is needed, as long as no autonomous operation over months is targeted.
- Fig. 12 is very small and difficult to read. Also, the information would be much more accessible (and interesting) in the form of vertical profiles instead of timeseries.

In summary, I very much like the profiling system presented by the authors and strongly encourage them to focus on this aspect, presenting its performance and limitations in an accessible way to the readers and the community. I think that the paper will gain in attraction, if shorter and more focused on the vertical profiling capability.

---

## Author Comment (AC1)

**A reply to https://doi.org/10.5194/amt-2022-208-RC1**

**Original referee comments in blue, with author response in black.**

This paper presents a new compact packing system for a commercial cavity ring-down laser spectrometer that is intended for in-situ deployments in harsh (here cold, polar) environments. The manuscript also presents a few additional modules such as a cold trap module and a profiling arm. While I find the paper well-written and very interesting to read, I have a few fundamental critiques in particular with respect to the structure and focus of the paper that the authors should reflect upon and address before acceptance of the manuscript.

**1) Technical innovation and significance:** While I really like the level of technical detail and completeness of the description of the ISE-CUBE, I do not fully understand why it stands in the center of a paper publication. Dozens of previous scientific investigations have been conducted in different in-situ installations of cavity ring-down spectrometers in containers, cars, ULM, ventilated aluminum housings, tents or aircraft racks. All these deployments were done in such a way as to address the scientific question at hand in the best possible way. The ISE-CUBE seems a useful packing method for exactly the chosen deployment: namely near-surface profiling of stable water isotope gradients in cold environments. But already in the midlatitudes and especially in the tropics the chosen setup would not work due to overheating. In my view the technically relevant and innovative part of this study is not the CUBE but the profiling arm, which however is only very sparsely addressed. Therefore, in my view the profiling arm should stand in the center of the story framing. The full use of the compact packing provided by the ISE-CUBE only becomes obvious, when combined with the profiling arm. The authors should seriously reconsider their storyline, provide a better literature-overview of existing studies with different in-situ deployments, and justify why such a detailed presentation of a very specific packing is useful to the community. To me the fact that the system is not autonomous in terms of power use is a big drawback and doesn't make the system so much more flexible than a sheltered installation with a long inlet-line.

We agree that a reframing of the manuscript as you propose would be very beneficial. Therefore, in the revised manuscript, rather than present our work as a "field **deployment** system" with the focus on the primary modules (analyzer and pump), we will shift the focus to an entire "field **profiling** system", where the profiling modules feature on an equal level as the primary modules. We maintain the equal importance of all three of these modules. As will be addressed in a comment below, the cold trapping module will remain as an optional expansion module. We will also expand our description to include an alternate profiling arm frame that we used to make profiles at the fjord. A possible revised title would be "*A modular field system for vertical profiling of in-situ vapor in harsh environments using cavity ring-down spectroscopy*"

**2) Motivation for a profiling system of the near-surface profiles within 2 m above the surface:**
- As mentioned above I really like the profiling arm and think that this is a clear innovation and add-on to the current state of the art in the isotope literature. It also has in my view clear potential for scientifically relevant investigations. The authors should mention these in the introduction more explicitly: why is it important to measure near-surface humidity/isotope and potential other trace gas gradients up to 2 m height?

The primary science aim driving the development and construction of the system was on the near-surface interaction and exchange between the surface and overlying air during processes such as evaporation and condensation. Our ultimate goal is to use this system to better understand the isotopic fractionation across the surface-atmosphere interface in these situations from profile measurements. We will better communicate this motivation in the introduction, since as the manuscript now stands, this is only briefly mentioned.

- Normally bulk fluxes are computed using measurements over about 10 m depth near the surface, why are the authors interested in the lowest 2 m?

Scalar surface fluxes can be measured at 2 m (Foken, 2008), and a lower measuring height produces a more localized footprint for our observations. As our focus is on the surface exchanges and given the generally very stable stratification in the Arctic surface layer during spring, we determined 2 m to be sufficient for our science aims mentioned above.

- The authors should highlight more clearly in the introduction why in a polar environment it is of utmost importance to have short inlet lines (due to strong interactions with the tubing walls at low concentrations, longer response times, lower precision at low concentrations).

We will make sure to clarify this point in the revised manuscript, especially as it directly ties to our design motivation for the analyzer and pump modules

- What makes a profiling arm with free choice of sampling height more valuable than a setup with a manifold and inlets at discrete heights? This is an essential argument for the profiling arm and should come very early in the manuscript. It is now mentioned only at the very end at L. 465.

We briefly touch upon the additional advantage that the arm has over tidal waters in L.483, but we will expand upon both points earlier in the manuscript.

**3) Section 4.1 & 4.2:** this section is much too detailed: 7 pages to state that the measurements were essentially unaffected by the harsh environmental conditions seems exaggerated to me. I am conscious of the effort that the authors put into the data analysis to come to that conclusion (stated at L. 374-376) and I fully acknowledge that this effort is worthwhile. Figures 5 to 7 with respective tables and shortened text would make an excellent supplement. But the information given in the paper should be succinct. The DAS temperature is not that relevant for the measurements, much rather the cavity temperature and pressure should be kept stable (this can be summarized in a few sentences). The WLM discussion in Section 4.1.3 remains inconclusive to me. The importance of the air prewarming by using the exhaust of the pump module can be mentioned in the methods section. A maximum 1-page summary of these results putting forward mainly the results of Section 4.2 (L. 399-402) with Fig. 10 should be sufficient to describe the main results and keep the reader's attention focused.

We appreciate the recognition of the time and effort gone into our analysis of the CRDS performance, though we ultimately agree that these sections are too intricately (and exhaustively) presented. In accordance with your previous suggestion, and our proposed reframing of the manuscript as a field profiling system, the main focus will be on the integrity and reliability of the profiles, from which the performance of the CRDS naturally follows. We will significantly shorten these sections to better respect the attention of the reader.

**4) Cold trap module:** This is an interesting adaptation of the Peters & Yakir 2010 system. However, if no data from this system is shown and compared to the CRDS data, then

this part should be left away. Currently, this part of the paper is difficult to assess without data.

While our cold trap samples are not quantitatively compared to the CRDS isotope data, we do assert that our qualitative assessment of the setup (ie. the need for a different type of collection vial) is sufficient grounds for inclusion. We've taken a system proven by Peters and Yakir (2010), and attempted to apply it to a much different environment than they tested in. In the revised manuscript, we will present some data for an initial assessment of the system performance.

**5) Field calibration expansion module:** I do not understand why a calibration module is useful for such a deployment, which needs manual handling of the system anyways. Recent studies have shown that CRDS systems operate reliably over the timescale of several days with minimal drift (without calibration), such that a calibration every few days (1-3) is entirely sufficient and can be done in shielded temperature regulated conditions. See also the statement of the authors themselves at L. 220.

In the context of our particular profiling deployment, we agree that the necessity of a field calibration unit is limited, hence why we deemed it sufficient to calibrate in the lab before and after deployment. However, in the larger context of our presentation as a deployment system that might be used for longer periods, we thought it important to emphasise the necessity of proper calibrations. In this regard, we will remove Section 2.6, as it is not relevant to our particular deployment. Though we will keep our mention of it in the Outlook section, as anyone considering to deploy for longer periods of time should incorporate a calibration system into the setup.

**6) Profiling module performance:** as mentioned above, I think that the real innovation of this paper is this profiling arm, which also makes the need for a low-footprint and modular box clear to keep the length of the inlet line at a minimum. Unfortunately, the authors put much more effort in sections 4.1 and 4.2 than in the key sections 4.4 and 4.5.
- I recommend restructuring these sections and showing more results on this essential part. In my view L. 421 to 428 should be in the methods.

These will be moved to the methods section.

- The response time of the system & precision at the encountered concentration should inform about the ideal length of the measurement periods at a given height. This point should be discussed. Is 30 s averaging ideal also from a signal-to-noise ratio perspective? Or should it be longer?

In our revised manuscript, we will discuss the inlet response time in more detail. And as we will be discussing the profiling capabilities in more detail, we will discuss our findings on the minimum time left at each height.

- The temperature sensor calibration is a good thing to do, but a Supplement figure would be sufficient.

Figure 11 is not a calibration, but a comparison with the nearby AWI station. However we will consider moving it to an Appendix.

- Fig. 12 is (one of) the most interesting figures (together with the very nice technical drawings in Figs. 1-3) but it is difficult to read because it is shown as a time series (and too small with many panels). A profiling arm allows to measure profiles, so why not show profiles? When looking at Figure 12 and considering the main aim of the paper (providing a modular system that is able to measure near-surface isotope and trace gas profiles in cold environments) then I wonder: can the proposed setup resolve the

vertical gradients given the uncertainty of the measurements at these low concentrations? The authors should show the vertical profiles under different conditions including the total uncertainty of the measurements and discuss this very important question. Also, in addition to the isotope, temperature and wind information they should add the water vapour mixing ratio and dexcess.

We agree with your suggestions. Especially in regards to your last point, we will re-make Figure 12. As we've explained previously, under our original manuscript structure, presenting the measurements from both the Analyser and the Profiling module as a time series made the most sense, as both data streams are fundamentally just that. In the new framing, we will present the measurements from the CRDS as vertical profiles. We will also change the day that is presented from 28 Feb to 9 Mar, which is more suitable to illustrate the profiling operation.

**Detailed comments:**
I refrain from a detailed list of language and technical comments here, given my advice above for fundamental reorganization of the paper. I however chose to list a few points that need clarification in the text:can the proposed setup resolve the

vertical gradients given the uncertainty of the measurements at these low

concentrations?
- L. 15: which processes? Those relating to fluxes?

This should read "these processes", referring to the exchange processes of the previous sentence.

- L. 17: during stable stratification -> really only then? I can imagine many situations in which the stratification is not stable and in which near-surface measurements would be very useful.

This will be changed to "Near surface (<2m) gradients over the snowpack can be strengthened significantly due to the stable stratification that often occurs in these regions, and ultimately govern the fluxes of …"

- L. 20: "disentangling water vapor of different origin and undergoing different" processes -> do you want to disentangle the water vapor? Or the different sources of the water vapour?

The isotopic composition can be used to disentangle the moisture origin. We will clarify this in our revisions.

- L. 23: did Steen-Larsen et al. 2013 investigate moisture sources? Or airmass origin? Maybe choose a different reference here. Also Sodemann 2017 is a proposal that is not accessible online and not a document that I would expect to be referenced in a peer-review paper. Maybe Sodemann et al. 2017 is meant?

True, Steen-Larson et al. 2013 looked into arimass origin, and a different reference would likely stand better.

Yes, thank you and well spotted. This was likely from a mixup between Sodemann2017**a** and Sodemann2017**b** in the citation software.

- L. 26: remove "or so" (spoken language)

We will change this to "Over the last few decades…"

- L. 30: what is the advantage of an in-situ system such as ISE-CUBE-profiling-arm over

a line with a manifold? Please be more explicit. This touches upon the key innovation
of this work.

The arm can be set at any height in an observational range (~2 m), and can therefore generate a profile observed across a multitude of levels (N). These multiple levels better capture the near-surface profile, which may have a non-linear shape. Any similar approach with a manifold system would involve N-times the amount of inlet lines. Additionally, the flexibility and the distance sensing capabilities of the arm enables us to observe over a changing reference level (i.e. tidal waters). We will be sure to expand upon these advantages in the revised manuscript.

- L. 51: Wall effects -> indeed very important and how is that addressed? How long are the response times of the system? This is important for the profiling strategy (i.e. how long does the arm stay at a given elevation)

Our efforts to mitigate the wall effects (stainless steel tubing, heated lines, minimal length) are brought up in Section 2.5, albeit insufficiently given our proposed focus change. Response times will be elaborated upon in Section 4.4.

- L. 101: stand-alone field operation -> no power (how much in total?) is needed

Stand-alone refers to the fact these two modules (Analyser and Pump) can stand on their own, apart from the other two modules, not stand-alone from grid power. The sentence will be clarified.

- L. 169: I would say this is a typical example of an unstable situation at least over the open ocean.

Agreed. When referring to stable conditions/startifications in the manuscript, it pertains to the conditions over the snow.

- L. 190: An overview … is given.

We will change accordingly.

- L. 204: if that is a central tool to this publication it should be made available online along with the data

We will expand our description of the routine to further detail the processing that the script does.

- Section 3.4: I have difficulty assessing if the comparison dataset from the lab is an adequate one to use. Is the amount of data (sample size) and sampling frequency comparable to what was used in the field? The description is a bit vague in this respect.

Our lab benchmark is approximately 5 times larger than the field dataset. In the lab, the analyser sampled at 1 Hz, while in the field the analyser had a sampling rate of 4 Hz. However the comparisons done between the two environments are both at 1 Hz.

- L. 387: the fact that the vials have to be manually changed in the cold trap module should be mentioned in the methods.

We will include this detail in the revised manuscript.

- L. 353: I cannot follow the argument why the only slightly larger RS at low water vapour mixing ratios in the field necessarily implies less accurate measurements in the Field.

As per Johnson and Rella (2017), the residual represents the difference between modelled spectrum and the observed, best-fit spectrum measured by the analyser.  The best-fit routine makes adjustments to free parameters such as the amount of the various isotopologues to the

observed spectrum, in order to minimize this residual. With a larger residual at a particular humidity level, measurements can be considered as less precise. See our comment below for more details on our proposed revisions for Section 4.1.3 and Fig 8.

- L. 360-362: so then why such a detailed discussion of these metrics?

In our revised manuscript, we will introduce the Zeppelin Observatory dataset in Section 3, as its own reference period for the WLM. Therefore, we will forego the discussion of the Laboratory WLM metrics; Fig 8 will essentially be replaced by Fig D1. This will also work towards shortening the manuscript.

- L. 416-419: I don't understand why the authors introduce the cold trap module if they don't use its data. That makes it difficult to assess if the system is fulfilling its purpose.

As mentioned above, we will be including cold trap analysis of select samples in the revised manuscript.

- Fig. 11 should go to the supplement.

We will move this figure to the Appendix

- L. 424: how were these response times estimated, to me the averaging intervals are also a key factor for optimizing the signal-to-noise ratio and obtaining the best possible precision.

We will discuss the inlet response time in more detail in our revised manuscript.

- L. 449: "strongly stable", this is even an inversion

Yes, other nearby instrumentation documented a persistent near-surface inversion throughout much of the time at Snow. However, this profiling example will change in the revised manuscript.

- L. 454: which isotope gradients do you mean here?

$\delta^{18}O$ and $\delta D$. We will state this in the text.

- L. 458: we captured d18O and dD (leave away "isotope signatures" of, that is a repetition)

We will change accordingly

- L. 462: what does "the temperature gradient… converged" mean here?

The temperature gradient between lowermost and mid levels weakened and approached the value of the gradient between mid and uppermost. In Fig 12e, the black dotted line converged towards the black dashed line.

- L. 463: not shown -> but that would be very interesting!

We are currently preparing a manuscript for Earth System Science Data.

- L. 463: this is very important and should be mentioned in the introduction as a motivation for the ISE-CUBE with profiling arm system.

We will ensure that we emphasize this strength in the revised manuscript.

- L. 483: "Flexibility of the measurement's height… with strong tides" interesting, but I missed that argument in the results part of the manuscript

As mentioned above, we will introduce this design consideration earlier in the manuscript.

- L. 486-491: the authors should compare the cold trap sampling to the CRDS measurements or leave it away.

       As mentioned above, we will consider including comparative data from our cold trap sample analysis in the manuscript.

- L. 492-498: As mentioned above, I do not understand why this is needed, as long as no autonomous operation over months is targeted.

       Agreed, though we will keep mention of this potential module here in the Outlook. We feel that we would be remiss to not mention the importance of proper calibrations, for which an in-situ unit would be necessary in deployments approaching 3+ weeks (Leroy-Dos Santos, 2021).

- Fig. 12 is very small and difficult to read. Also, the information would be much more accessible (and interesting) in the form of vertical profiles instead of timeseries.

       Yes, we will be changing Fig 12 to show profiles alongside the timeseries. The text of the figure will also be larger.

In summary, I very much like the profiling system presented by the authors and strongly encourage them to focus on this aspect, presenting its performance and limitations in an accessible way to the readers and the community. I think that the paper will gain in attraction, if shorter and more focused on the vertical profiling capability.

References

Foken, T.: Specifics of the Near-Surface Turbulence. In: Micrometeorology. Springer, Berlin, Heidelberg. https://doi.org/10.1007/978-3-540-74666-9_3, 2008.

Johnson, J. E. and Rella, C. W.: Effects of variation in background mixing ratios of N2, O2, and Ar on the measurement of δ18O–H2O and δ2H–H2O values by cavity ring-down spectroscopy, Atmospheric Measurement Techniques, 10, 3073–3091, https://doi.org/10.5194/amt-10-3073-2017, 2017.

Leroy-Dos Santos, C., Casado, M., Prié, F., Jossoud, O., Kerstel, E., Farradèche, M., Kassi, S., Fourré, E., and Landais, A.: A dedicated robust instrument for water vapor generation at low humidity for use with a laser water isotope analyzer in cold and dry polar regions, Atmospheric Measurement Techniques, 14, 2907–2918, https://doi.org/10.5194/amt-14-2907-2021, 2021.

---

## Author Comment (AC2)

**A reply to https://doi.org/10.5194/amt-2022-208-RC2**

**Original referee comments in blue, with author response in black.**

This manuscript describes a new modular box enclosure called ISE-CUBE that can be used to deploy water vapor isotopic analyzers and water vapor isotopic cold-trap systems in the field under extreme cold-weather conditions. The manuscript provides a short description of the enclosure and subsequently evaluates the isotopic analyzer's housekeeping variables from a two-week winter deployment in Svalbard. The housekeeping data suggest the analyzer is able to maintain satisfactory ranges for its Data Acquisition System temperature, its cavity temperature and pressure, and its warm box temperature. The analyzer's water isotopic measurement precision in the field is also comparable to its measurement precision while sampling calibration gas in a laboratory setting.

The manuscript also describes an optional "profiling module" for ISE-CUBE, consisting of a tripod with an articulating measurement arm, that can be used to position a heated inlet line for the isotopic analyzer anywhere from 4 to 205 cm above the ground surface. A 90-minute window of data is presented that shows water vapor concentrations and isotope ratios from six height levels within the articulating arm's 2-m range. The paper argues that the articulating arm provides a means to resolve and study the water and isotopic gradients closest to the surface, although doing so requires repositioning the inlet height via a manual pulley every few minutes.

**Comments**

Where this paper really advances our measurement abilities is in the design and presentation of the box enclosure for the isotopic analyzer and cold trap; yet most of these details are in the Supplemental material instead of in the main paper. I would recommend revisiting Figures 1-2 and using these to convey specific details about the box connections and tubing materials, something more akin to the Connectors Template in the SI. As currently presented, Figure 1a is simply too dark to make out details, and Figure 1b requires more detail and explanation. For example, what are the "power out" and "data" ports used for? Where are the fan inlets and exhaust ports? Where do the boxes connect to one another? Is the CRDS inlet unheated after the check valve? Which lines are PTFE and which SS? In addition, the main text mentions components such as an "adapter," an "exterior inlet bulkhead," "incoming ventilation tubing," and "manifold tubing." Can these be labeled on Figs 1-2? The list of components in Appendix A is fantastic. Consider also a corresponding diagram (again, like the Connectors Template) that shows where all these components go and telling readers how many of each part are required to replicate the system. Another way to think about this: what would a purchase list look like?

We agree that the manuscript as a whole would be enhanced by annotated diagrams. In the revised manuscript, we will therefore include remakes of Figs 1 & 2 that take the reviewer's concern into account. However, upon the recommendation of Referee #1, the manuscript focus will shift slightly to be a presentation of a "field **profiling** system" rather than as a "field **deployment** system". Therefore, we will revise the manuscript to elevate the role of the profiling arm. Thereby, the cold trapping module will remain as an expansion to the profiling system, as mentioned in one of the comments below. In this context, we feel as

though some of any additional diagrams might still fit best in the supplemental, or perhaps as an appendix, with the majority of the manuscript text being devoted to proving the validity and reliability of the profiling operations. We will expand the Supplemental material to include more building details, including estimates for costs, and potential cost cutting measures.

It would be helpful if the manuscript discussed the relevance of the ISE-CUBE enclosure to the wider measurement community. Much of the manuscript is specific to the deployment of a Picarro CRDS water vapor isotopic analyzer. Would ISE-CUBE work for other types of isotopic analyzers? If the enclosure is specific to the size and shape of the Picarro systems, could ISE-CUBE work for other gas-phase Picarro analyzers? Moreover, based on the short two-week deployment in Svalbard, is there any sense whether ISE-CUBE could last for longer periods for unattended measurements?

Yes, we had the potential of other analyzers in mind while writing the manuscript. As the manuscript stands now, we only briefly mention this in the Introduction and in L.508 in the Outlook. But we believe that this system could be used with other CRDS analysers manufactured by Picarro; we will clarify and expand on this in the manuscript.

Whether or not the system could be deployed for longer periods is a nuanced and detailed point, which we now see would fit very well in the manuscript. In the revised manuscript, we will include a section presenting the limitations and strengths of the system more clearly.

On a related note, the Data Processing section (Sect. 3.2, including Table 1) presents ISE-CUBE as producing three data streams generally, but these three streams are specific to the way the modular system was set up for testing during ISLAS2020. It would be helpful if the paper distinguished more carefully which aspects of the design are generic and applicable broadly vs. specific to the test case configuration.

With our new focus, the data streams generated by the Analyzer and Profiling modules are the basic outputs, while the Cold Trap output would be specific to our deployment. We will rephrase this paragraph to clarify.

The enclosure is presented as novel, in part, for minimizing disturbance to the environmental flow, but I think this claim might be overreaching, since most ground-based installations are designed to minimize flow disturbance (e.g. flux towers). The real draw of the enclosure in my mind is the ability to deploy a water vapor isotopic analyzer in an environment with minimal infrastructure support (e.g. nothing more than a power drop) and/or to reduce the length of inlet lines and thus measurement hysteresis.

We take your point that such a claim can be overreaching, especially with the current structure of the manuscript. However when considered and presented together as a near-surface profiling system, we believe the value and novelty of the system's compactness is emphasised.

To evaluate ISE-CUBE, the water vapor isotopic analyzer's performance in the field is compared to its performance in the laboratory. The intention is to compare two distinct environmental settings. However, there is another relevant difference that needs to be communicated more transparently: in the laboratory, the analyzer samples reference gas continuously, whereas in the field, the analyzer is measuring real variability related to the environment. I would not be surprised if this difference in sampled air causes the differences in humidity-binned standard deviations presented in Fig. 10 or results in the differences in

spectral-fit residuals (RS) presented in Fig. 8. The paper concludes that the field data are "marginally less precise," but, again, I wonder if this is not just a reflection of the environmental air. Would one reach the same conclusion if the analyzer were measuring reference gas while deployed as part of ISE-CUBE in the Arctic?

This is a very interesting point, though since we didn't have a way to make a direct analog to the lab (i.e. a constant vapor stream of standard), it is ultimately difficult to determine. We will consider to reformulate our conclusions as "marginally less precise **than optimal measurement conditions**". We will mention this possibility when discussing Fig 8 & 10.

While I'm not sure it is necessary for the point the paper is trying to make, it would be awfully interesting to see how the isotopic analyzer and cold trap compare during the ISLAS2020 deployment. Such a comparison could provide some indication of the accuracy of the isotopic analyzer when deployed with ISE-CUBE.

We will include a brief evaluation from two cold trap samples in the manuscript. However, we don't believe that the system is mature enough such that it can currently indicate the accuracy of the CRDS. The design is proven in Peters and Yakir (2010), but we were deploying it for the first time under very different conditions. Regardless, this module will stay categorised as an "expansion".

Lastly, for Fig. 12, it appears there are environmental data missing during the period highlighted in the text (9:07, onwards). In addition, it would be helpful to know, are these data from the AWS? And can the figure be made larger?

These periods are, unfortunately, actual gaps in the data. However, in response to Referee #1, Figure 12 will be significantly modified in our revisions. Since the system is being put forward as a profiling system, it naturally follows to show the vertical profiles. Therefore, we will change the day we show (to 9 Mar 2020) in order to best present an example of an actual profile. Right now, the particular day shown in the manuscript was chosen as an illustration of its iterative height capability (after around 9:20), which is well visualised in a time series.

**Overall, the paper is very clearly written; however, a few minor comments on presentation are provided below:**

L 15 - perhaps "components" instead of "compartments"

Changed to "reservoirs in the Earth System."

L 29-32 is a bit awkward and could be presented more clearly

Under our proposed reframing, we will expand upon the limitations of fixed height inlet systems, and the benefits of a continuous inlet line.

L 39 and elsewhere - "pneumatically" might be the wrong word as this implies compressed air

We will find a different term to avoid any confusion.

L 165 - is there a reference for ISLAS2020?

The data paper for this campaign is in preparation for Earth System Science Data.

L 177 - Does the ISLAS2020 data span 21 Feb to 14 Mar?

It does, though part of that time (29 Feb to 3 Mar) had our instrument installed up at Zeppelin Observatory (472 m ASL). The remaining days were dedicated to calibration and maintenance inside the Marine Laboratory.

L 187 - "reliable" seems like the wrong word for what is intended
Changed to "comprehensive"

L 219 requires clarification
Each of the 17 individual calibrations had standard deviations equal to or slightly larger than the standard deviation found across the mean of all calibrations. We will make sure to clarify this in the text.

L 239 and elsewhere - "minutely" means "meticulously." I think the paper intends to say "1 minute"
Yes, we will change this accordingly.

L 375 - since "field" and "laboratory" have specific meanings, I would use "remote observatory" or some other phrase here
It was an intentional blending of the two terms, but we see how it can be confusing, especially as we've already established it as an "observatory". This line will read "same quality as a remote observatory"

Fig. C1 - caption says observatory margins are for the same "time" but not the same dates, right? Perhaps clarify.
Yes, the sentence is confusing. The light grey shading indicates the "same" 2nd and 98th percentiles, *from* the time period when we were measuring in Zeppelin Observatory. We will change the sentence to clarify this distinction.

L 521 - says "could be" but should it say "is" or does it really mean "might be comparable" (as in it's unknown)?
More so the former, since we do know it. The $T_{WB}$ recovery times were longer during the longer profiling periods, as compared to briefer routine site checks. But the magnitude of the initial $T_{WB}$ dip/spike *could be* just as large during a site check, as during a profiling period. We will rephrase the sentence to clarify this.

Fig. D1 seems to be missing the gray shading mentioned in the caption
The grey shading is similar to the width of the dotted line, and is almost invisible underneath the dotted line; we will mention this in the figure caption.

References

Peters, L. I. and Yakir, D.: A rapid method for the sampling of atmospheric water vapour for isotopic analysis, Rapid Communications in Mass Spectrometry, 24, 103–108, https://doi.org/10.1002/rcm.4359, 2010.

---

## Author Comment (AC3)

**A reply to https://doi.org/10.5194/amt-2022-208-RC3**

**Original referee comments in blue, with author response in black.**

General comments:

I found this to be a clearly written paper, well-structured and systematic. It is a little long for the content so an attempt to shorten some sections would be welcome.

We will reduce the length of the manuscript, in particular in Section 4.1. Additionally, upon the recommendation of Referee #1, the manuscript focus will shift slightly to be a presentation of a "field **profiling** system" rather than as a "field **deployment** system". In this regard, we will also shift the focus of the discussion to the reliability of the profiles, from which the performance of the CRDS analyzer naturally follows. We will also now be including details on an alternative profiling frame that we used while at the Fjord measurement site.

It provides a detailed account of the design and testing of a well engineered set of enclosures for the isotope analyser and some peripheral equipment. It looks to be quite costly - please provide the cost of the parts to build this. Many users of these instruments would look for low-cost solutions to building enclosures and peripherals and in many cases sensible cost cutting can be made without substantially affecting performance.

Upon the recommendation of Referee #2, we will be providing a detailed purchase list of more components in the Supplemental "builder's guide", including the amount of each item for each module. We will also include the approximate cost of each item. We do already put forward some alternative connectors in the Supplemental.

The paper would benefit from a better justification of the need for this design, i.e the advantages of this design compared to other solutions to obtain such data. For example, given that a power source must be nearby (presumably within a few 10's of meters) why couldn't instruments be installed indoors with inlet tubing to the outdoors? Long inlet tubings are routinely used on tall masts with an appropriate pump rate? Are there particular problems regarding lag time? Or memory effects? Or disturbance of the air stratification? Why couldn't an existing mast be used with multiple inlet heights and the use of a manifold? A nearby EC tower is mentioned in the paper.

We see these questions underline the value of shifting focus in the revised manuscript, as recommended by Reviewer #1. Thereby, we will specifically highlight our aim of making in-situ profiles of the near-surface exchange between the surface and overlying air. Therefore, we required response time faster than provided by long inlet lines. At the Snow site, we tied into the power supply of the nearby EC site, but the nearest building that could adequately house the Picarro was on the order of 300 m away, which we deemed to not be a viable option. As a result, it follows that we needed a Picarro housing that would minimize disturbance to the measurement site (albeit not eliminate as mentioned in a comment below), while being as close as possible. In regards to the advantages that the arm provided us over a manifold setup, we briefly touch upon these on L.464 & 483, but we will expand upon both points earlier in the revised manuscript.

The publication could be seen as premature given the preliminary and incomplete parts (cold trap, pivot arm and standard gas supply module). However, it may be that the authors had limited opportunities to test the device in this remote location.

While we don't agree that the manuscript is premature, we do recognize that our initial scope as an all-around Picarro deployment platform might be seen as overreaching. However, we do assert that the capability that the system provides for making near-surface profiles in polar environments is novel and worthy of a publication.

The paper is mainly an account of environmental measurements (T and P, spectral characteristics etc) of the analytical instrument when placed inside the housing under cold and windy conditions, rather than a more complete account of a test of actual isotopic measurements. Ultimately the most critical isotope test, i.e. a comparison of one or more constant gas supplies both in the field and the laboratory, could not be carried out due to the lack of a working standards gas supply module. Also, there seems to have been a missed opportunity to compare the real time isotope data with isotope data obtained from the cold trap sampling, why wasn't this done (or presented) here?

Yes, we do agree that exposing the instrument to standard gas while deployed would provide for a definitive proofing of the system. However, no such standard gas supply module was available for field deployment under such extreme ambient conditions. Therefore, we instead focus on the analyzer metrics related to spectroscopy, the most critical being the cavity temperature and pressure, which we do readily have. Then, by comparing these metrics to analogous operation inside a controlled laboratory environment, we can discuss the impact the deployment made on the isotopic measurements.

We don't believe that the cold trap can currently provide fair insight into the accuracy of the CRDS. The design is proven in Peters and Yakir (2010), but we are deploying it under very different conditions as a test. In that sense, we will revise our statement on P.5 L.105 to put forward that discrete sample analysis gives the option for calibration, rather than with the intention of using our particular samples for that purpose. We will edit this section to better reflect this intention. In the revised manuscript, we will be including analysis of select cold trap samples for a first comparison. Thereby, we will maintain the cold trapping module as an "expansion" to the overall profiling system.

Given the anticipated degradation of isotope ratio precision at low H2O mixing ratios (typical of polar regions), and the possible instrument drift due to the changing environmental conditions, it would be important to check standard gasses a regular interval (likely several times daily) in an actual measuring campaign, hence the need for a standard gas supply device. As mentioned, this has not been demonstrated in the current manuscript.

While the low concentration does negatively impact the measurement precision of the instrument, we anticipate minimal drift in the instrument over our deployment timespan. Older models of Picarro instruments (L1102i) have shown a much stronger tendency to drift during previous studies (Steen-Larsen et al., 2013). Leroy-Dos Santos (2021) has shown no measurable drift in an L2130i over a 3-week period at quite low humidities. However, longer (profiling) deployments with our system that would start to approach this timescale would require an in-field calibration system, which we mention in Section 5. We will further expand on this discussion in the revised manuscript.

A more thorough explanation of the stratified air column data obtained would benefit the paper as it would demonstrate the useful application of the enclosure and pivot arm. The need for an operator to use the pivot arm seems to risk disturbing the stratified air column, depending on wind direction (creating turbulence)

Yes, we agree, which is why the manuscript will now shift focus. For this reason, Figure 12 will change to show a profiling period from 9 March.

Yes, the potential for site disturbance is an argument for the remote/automated operation of the arm, which would be great to develop further. However our deployment orientation at the Snow site ensured that we were downwind of the arm for the vast majority of the deployment period.

Specific comments:

P1 L12: I don't think you can claim it would be satisfactory in all environments, e.g. in the tropics, overheating, condensation of humidity may be a different challenge, in environments with high day – night temperature contrast drift may be problematic

We did not intend for this sentence to make such a claim. We will rephrase it to more accurately describe the potential we see in the system.

P2 L41: What are the conventional approaches?

Conventional deployments inside tents or person-sized enclosures, which are on the order of 5 cubic m or larger. We will make this comparison clear in the text.

P2 L49: But you have a nearby power source so why not house the instrument there?

The nearby power source is the EC mast, which does not have any adequate housing for the Picarro.

P2 L52: But long inlet lines (fluorinated plastics) are routinely used in tall towers

As mentioned in a comment above, our aim was to minimize instrument response time, which involved shorter lines.

P3 L64: Is it pneumatic? Maybe just gas or airtight connectors?

Yes, pneumatic isn't quite the right word. We will find a different term.

P4 L85: Isn't a lower flow rate preferable to increase precision in dry air? Especially since there's a separate high-flow pump to deliver the air sample close to the inlet?

This is a very interesting point, and one that we did not consider in our planning. Our aim was to minimize response time, which meant increasing flow. We will be mentioning this potential setup in our the newly added Discussion section.

P5 L106: So why wasn't this done when there was no available standard gas module available?

As we mention above, we will rephrase this paragraph to better communicate our expectations from the cold trapping module.

P6 L144: Does the presence of an operator disturb the stratification?

There was no discernable disruption of the stratification at the sampling site. Additionally, we positioned the operator to be downstream of the sampling inlet.

P7 L161: This would have been the most complete test of the system

As in a comment above, yes, it would have been the most complete test. However, in its absence, we maintain that the analyser diagnostics provide adequate proof of operation.

P10 L211: Please state if these are liquid or vapour values

These are values of the liquid standards used in the Picarro SDM.

**P10 L220: This sentence is unclear – does it mean that measurements were within +/- 1 sigma?**

The total calculated measurement drift during the campaign was smaller than the standard deviation of the 17 calibrations. This standard deviation across the 17 calibrations was also similar in size to the standard deviation of any individual calibration.

**P10 L229: Not sure this is correct, there are numerous field applications documented on the Picarro web site.**

Very much agreed, however the form of the CRDS analyser is *designed* for use on a laboratory bench, which is where it should be operating at its best. We certainly did not mean to imply that no one else had used Picarro's in the field. This possible misunderstanding is why we have decided to switch focus to emphasize the system's holistic profiling capacity.

**P11 L247: Don't think 'minutely' is a word (?)**

Quite correct; we will fix this phrasing at this and all other points in the manuscript

**P11 L259: Please specify where DAS is measured**

The DAS temperature is measured inside the exterior housing of the analyzer, on an internal circuit board. It is taken to represent the overall temperature of the analyser circuitry (not including components in controlled casings).

**P16 L355: As mentioned above, wouldn't the normal low flow setup have been preferable?**

As mentioned above, we will be discussing this possible setup in the newly added Discussion section.

**P22 L454: add 'for d18) and dD, respectively**

Yes, we will change this accordingly.

References

Leroy-Dos Santos, C., Casado, M., Prié, F., Jossoud, O., Kerstel, E., Farradèche, M., Kassi, S., Fourré, E., and Landais, A.: A dedicated robust instrument for water vapor generation at low humidity for use with a laser water isotope analyzer in cold and dry polar regions, Atmospheric Measurement Techniques, 14, 2907–2918, https://doi.org/10.5194/amt-14-2907-2021, 2021.

Peters, L. I. and Yakir, D.: A rapid method for the sampling of atmospheric water vapour for isotopic analysis, Rapid Communications in Mass Spectrometry, 24, 103–108, https://doi.org/10.1002/rcm.4359, 2010.

Steen-Larsen, H. C., Johnsen, S. J., Masson-Delmotte, V., Stenni, B., Risi, C., Sodemann, H., Balslev-Clausen, D., Blunier, T., Dahl-Jensen, D., Ellehøj, M. D., Falourd, S., Grindsted, A., Gkinis, V., Jouzel, J., Popp, T., Sheldon, S., Simonsen, S. B., Sjolte, J., Steffensen, J. P., Sperlich, P., Sveinbjörnsdóttir, A. E., Vinther, B. M., and White, J. W.: Continuous monitoring of summer surface water vapor isotopic composition above the Greenland Ice Sheet, Atmospheric Chemistry and Physics, 13, 4815–4828, https://doi.org/10.5194/acp-13-4815-2013, 2013.

---

## Author Response (AR1)

Dear Editor and Referees,

In reply to the comments of the referees, we have made substantial changes to the manuscript. This includes large portions that have moved or been removed in the work, including additional analysis. In addition, many figures have changed, including the order. We believe that we have been able to address all comments and questions by the referees, and we believe that the work has been improved. Below we provide a point by point response. Please find the tracked changes of the manuscript in an additional attachment.

The Authors

RE: **https://doi.org/10.5194/amt-2022-208-RC1**

**Original referee comments in blue, with author response in black.**

This paper presents a new compact packing system for a commercial cavity ring-down laser spectrometer that is intended for in-situ deployments in harsh (here cold, polar) environments. The manuscript also presents a few additional modules such as a cold trap module and a profiling arm. While I find the paper well-written and very interesting to read, I have a few fundamental critiques in particular with respect to the structure and focus of the paper that the authors should reflect upon and address before acceptance of the manuscript.

**1) Technical innovation and significance:** While I really like the level of technical detail and completeness of the description of the ISE-CUBE, I do not fully understand why it stands in the center of a paper publication. Dozens of previous scientific investigations have been conducted in different in-situ installations of cavity ring-down spectrometers in containers, cars, ULM, ventilated aluminum housings, tents or aircraft racks. All these deployments were done in such a way as to address the scientific question at hand in the best possible way. The ISE-CUBE seems a useful packing method for exactly the chosen deployment: namely near-surface profiling of stable water isotope gradients in cold environments. But already in the midlatitudes and especially in the tropics the chosen setup would not work due to overheating. In my view the technically relevant and innovative part of this study is not the CUBE but the profiling arm, which however is only very sparsely addressed. Therefore, in my view the profiling arm should stand in the center of the story framing. The full use of the compact packing provided by the ISE-CUBE only becomes obvious, when combined with the profiling arm. The authors should seriously reconsider their storyline, provide a better literature-overview of existing studies with different in-situ deployments, and justify why such a detailed presentation of a very specific packing is useful to the community. To me the fact that the system is not autonomous in terms of power use is a big drawback and doesn't make the system so much more flexible than a sheltered installation with a long inlet-line.

We have reframed the manuscript to present a field profiling system. The name of the manuscript has also changed to "*A modular field system for near-surface, vertical profiling of the atmospheric composition in harsh environments using cavity ring-down spectroscopy*"

**2) Motivation for a profiling system of the near-surface profiles within 2 m above the surface:**
- As mentioned above I really like the profiling arm and think that this is a clear innovation and add-on to the current state of the art in the isotope literature. It also has in my view clear potential for scientifically relevant investigations. The authors should mention these in the introduction more explicitly: why is it important to measure near-surface humidity/isotope and potential other trace gas gradients up to 2 m height?

We now introduce this importance in Section 1 Introduction.

- Normally bulk fluxes are computed using measurements over about 10 m depth near the surface, why are the authors interested in the lowest 2 m?

Now put forward in Section 1 Introduction L.13: "Near surface (<2 m) gradients of scalars can be strengthened significantly due to the stable stratification that often occurs in these regions

(Jocher et al., 2012; Zeeman et al., 2015) , which ultimately govern the fluxes of trace gases such as methane, carbon dioxide, and water vapor."

- The authors should highlight more clearly in the introduction why in a polar environment it is of utmost importance to have short inlet lines (due to strong interactions with the tubing walls at low concentrations, longer response times, lower precision at low concentrations).

    We now introduce this importance in Section 1 Introduction.

- What makes a profiling arm with free choice of sampling height more valuable than a setup with a manifold and inlets at discrete heights? This is an essential argument for the profiling arm and should come very early in the manuscript. It is now mentioned only at the very end at L. 465.

    We now put forward the advantages of the arm over a fixed-height inlet system in Section 5 Discussion, as well as some general shortcomings of fixed-height inlet systems in Section 1 Introduction.

**3) Section 4.1 & 4.2:** this section is much too detailed: 7 pages to state that the measurements were essentially unaffected by the harsh environmental conditions seems exaggerated to me. I am conscious of the effort that the authors put into the data analysis to come to that conclusion (stated at L. 374-376) and I fully acknowledge that this effort is worthwhile. Figures 5 to 7 with respective tables and shortened text would make an excellent supplement. But the information given in the paper should be succinct. The DAS temperature is not that relevant for the measurements, much rather the cavity temperature and pressure should be kept stable (this can be summarized in a few sentences). The WLM discussion in Section 4.1.3 remains inconclusive to me. The importance of the air prewarming by using the exhaust of the pump module can be mentioned in the methods section. A maximum 1-page summary of these results putting forward mainly the results of Section 4.2 (L. 399-402) with Fig. 10 should be sufficient to describe the main results and keep the reader's attention focused.

    Section 4.1 has been significantly shortened, including a more clear and concise presentation of the WLM results, which now includes further analysis.

**4) Cold trap module:** This is an interesting adaptation of the Peters & Yakir 2010 system. However, if no data from this system is shown and compared to the CRDS data, then this part should be left away. Currently, this part of the paper is difficult to assess without data.

    We now present two analysed samples for assessment in Section 4.4 Cold Trap module performance.

**5) Field calibration expansion module:** I do not understand why a calibration module is useful for such a deployment, which needs manual handling of the system anyways. Recent studies have shown that CRDS systems operate reliably over the timescale of several days with minimal drift (without calibration), such that a calibration every few days (1-3) is entirely sufficient and can be done in shielded temperature regulated conditions. See also the statement of the authors themselves at L. 220.

    We have removed Section 2.6 Field calibration expansion module (proposed), but have moved some of the points to Section 5 Discussion regarding longterm deployments.

**6) Profiling module performance:** as mentioned above, I think that the real innovation

of this paper is this profiling arm, which also makes the need for a low-footprint and modular box clear to keep the length of the inlet line at a minimum. Unfortunately, the authors put much more effort in sections 4.1 and 4.2 than in the key sections 4.4 and 4.5.

Analysis in this section is now much more comprehensive.

- I recommend restructuring these sections and showing more results on this essential part. In my view L. 421 to 428 should be in the methods.

These lines have not been moved to the methods but have been modified into a more detailed presentation of the inlet response. See also the answer to the next comment.

- The response time of the system & precision at the encountered concentration should inform about the ideal length of the measurement periods at a given height. This point should be discussed. Is 30 s averaging ideal also from a signal-to-noise ratio perspective? Or should it be longer?

We have now revised the Section 4.3 Profiling module performance to focus on the system's response and the necessary time at a given height.

- The temperature sensor calibration is a good thing to do, but a Supplement figure would be sufficient.

This temperature intercomparison has been moved to the Supplement

- Fig. 12 is (one of) the most interesting figures (together with the very nice technical drawings in Figs. 1-3) but it is difficult to read because it is shown as a time series (and too small with many panels). A profiling arm allows to measure profiles, so why not show profiles? When looking at Figure 12 and considering the main aim of the paper (providing a modular system that is able to measure near-surface isotope and trace gas profiles in cold environments) then I wonder: can the proposed setup resolve the vertical gradients given the uncertainty of the measurements at these low concentrations? The authors should show the vertical profiles under different conditions including the total uncertainty of the measurements and discuss this very important question. Also, in addition to the isotope, temperature and wind information they should add the water vapour mixing ratio and dexcess.

Figure 12 has been remade into Figure 10. General wind information is given in the text of Section 4.5 Example of a profiling operation, as it was from a single level.

**Detailed comments:**
I refrain from a detailed list of language and technical comments here, given my advice above for fundamental reorganization of the paper. I however chose to list a few points that need clarification in the text:

- L. 15: which processes? Those relating to fluxes?

L.12: Changed to "knowledge gaps on these processes"

- L. 17: during stable stratification -> really only then? I can imagine many situations in which the stratification is not stable and in which near-surface measurements would be very useful.

L.13: "Near surface (<2 m) gradients of scalars can be strengthened significantly due to the stable stratification that often occurs in these regions (Jocher et al., 2012; Zeeman et al., 2015) ,

which ultimately govern the fluxes of trace gases such as methane, carbon dioxide, and water vapor."

- L. 20: "disentangling water vapor of different origin and undergoing different" processes -> do you want to disentangle the water vapor? Or the different sources of the water vapour?
       We have removed this sentence.

- L. 23: did Steen-Larsen et al. 2013 investigate moisture sources? Or airmass origin? Maybe choose a different reference here. Also Sodemann 2017 is a proposal that is not accessible online and not a document that I would expect to be referenced in a peer-review paper. Maybe Sodemann et al. 2017 is meant?
       We have removed this sentence.

- L. 26: remove "or so" (spoken language)
       L.21: We have removed the time reference. "Laser spectroscopy has enabled…"

- L. 30: what is the advantage of an in-situ system such as ISE-CUBE-profiling-arm over a line with a manifold? Please be more explicit. This touches upon the key innovation of this work.
       L.36-39: We present limitations of multi-line manifolds.
       Most of the advantages (as well as limitations) of the arm are now put forward in Section 5 Discussion.

- L. 51: Wall effects -> indeed very important and how is that addressed? How long are the response times of the system? This is important for the profiling strategy (i.e. how long does the arm stay at a given elevation)
       L.36: "Therefore, short, heated inlet lines limit potential interactions between water vapor and the inner walls of tubing. The use of short inlet lines also promotes a faster response of the CRDS analyzer"
       L.113: "1/4 inch stainless steel tubing (Swagelok Inc.), heated to 60 °C with self-regulating heat trace cable (Thermon Inc.), and surrounded with 2 cm thick foam nitrile insulation."
       Response times will be elaborated upon in Sect. 4.3 Profiling module performance

- L. 101: stand-alone field operation -> no power (how much in total?) is needed
       L.105: "Together, the Analyzer module and the Pump module are the two essential modules for in-situ isotopic measurement of water vapor."

- L. 169: I would say this is a typical example of an unstable situation at least over the open ocean.
       Agreed. When referring to stable conditions/startifications in the manuscript, it pertains to the measuring conditions over the snow.

- L. 190: An overview … is given.
       L.190: "The ISE-CUBEs produce two main data streams, pertaining to the Analyzer, and Profiling modules, with each module internally recording its own respective stream. The Cold Trap expansion module does a similar task for its own data stream. An overview of the information contained in these data streams is given in Table 1."

- L. 204: if that is a central tool to this publication it should be made available online

along with the data
We now detail and list the different time resolutions used in the manuscript on L.204-209.

- Section 3.4: I have difficulty assessing if the comparison dataset from the lab is an
adequate one to use. Is the amount of data (sample size) and sampling frequency
comparable to what was used in the field? The description is a bit vague in this respect.
Our lab benchmark is approximately 5 times larger than the field dataset. In the lab, the
analyser sampled at 1 Hz, while in the field the analyser had a sampling rate of 4 Hz. However the
comparisons done between the two environments are both at 1 Hz. As mentioned above, these
details are given on L.204-209.

- L. 387: the fact that the vials have to be manually changed in the cold trap module
should be mentioned in the methods.
L.149 now reads "*After the sampling period is complete, the flow is shut off with the needle
valve (Figure 3, "D"), and the vial is manually removed, sealed, and stored until laboratory
analysis, which can be done from the same vial.*"

- L. 353: I cannot follow the argument why the only slightly larger RS at low water
vapour mixing ratios in the field necessarily implies less accurate measurements in the Field.
L.333-336: "While this indicates that the measurements in the field have the potential for
larger uncertainty, obtaining an exact quantification of uncertainty from this difference is non-trivial
and requires access to proprietary analyzer details. Therefore, we now proceed with an alternative
method to quantify the quality of the water isotope measurements from the ISE-CUBE system."

- L. 360-362: so then why such a detailed discussion of these metrics?
We have done further analysis with the laboratory benchmark, limiting comparison to times
when only synthetic air is being used. This makes the comparison much more pertinent.
Additionally, we now introduce the Zeppelin Observatory dataset in Section 3, as its own reference
period for the WLM.

- L. 416-419: I don't understand why the authors introduce the cold trap module if they
don't use its data. That makes it difficult to assess if the system is fulfilling its purpose.
As mentioned above, we now include two analysed samples for comparison in Sect.
4.4

- Fig. 11 should go to the supplement.
This figure has moved to the Supplemental Material.

- L. 424: how were these response times estimated, to me the averaging intervals are
also a key factor for optimizing the signal-to-noise ratio and obtaining the best
possible precision.
We present the inlet response in Section 4.3

- L. 449: "strongly stable", this is even an inversion
Section 4.5 Example of a profiling operation has changed substantially, including this
line.

- L. 454: which isotope gradients do you mean here?

Section 4.5 Example of a profiling operation has changed substantially, including this line.

- L. 458: we captured d18O and dD (leave away "isotope signatures" of, that is a repetition)
Section 4.5 Example of a profiling operation has changed substantially, including this line.

- L. 462: what does "the temperature gradient… converged" mean here?
Section 4.5 Example of a profiling operation has changed substantially, including this line.

- L. 463: not shown -> but that would be very interesting!
We are currently preparing a manuscript for Earth System Science Data.

- L. 463: this is very important and should be mentioned in the introduction as a motivation for the ISE-CUBE with profiling arm system.
We now mention this and other advantages in Section 1 as well as in Section 5 Discussion

- L. 483: "Flexibility of the measurement's height… with strong tides" interesting, but I missed that argument in the results part of the manuscript
Now mentioned in Section 5 Discussion

- L. 486-491: the authors should compare the cold trap sampling to the CRDS measurements or leave it away.
As mentioned above, we now include two analysed samples for comparison.

- L. 492-498: As mentioned above, I do not understand why this is needed, as long as no autonomous operation over months is targeted.
We move the discussion of deployment duration to Section 5 Discussion

- Fig. 12 is very small and difficult to read. Also, the information would be much more accessible (and interesting) in the form of vertical profiles instead of timeseries.
This figure (now Figure 10) has changed substantially, including larger text.

In summary, I very much like the profiling system presented by the authors and strongly encourage them to focus on this aspect, presenting its performance and limitations in an accessible way to the readers and the community. I think that the paper will gain in attraction, if shorter and more focused on the vertical profiling capability.

References

Jocher, G., Karner, F., Ritter, C., Neuber, R., Dethloff, K., Obleitner, F., Reuder, J., and Foken, T.: The Near-Surface Small-Scale Spatial and Temporal Variability of Sensible and Latent Heat Exchange in the Svalbard Region: A Case Study, ISRN Meteorology, 2012, 1–14, https://doi.org/10.5402/2012/357925, https://www.hindawi.com/journals/isrn/2012/357925/, 2012.

Zeeman, M. J., Selker, J. S., and Thomas, C. K.: Near-Surface Motion in the Nocturnal, Stable Boundary Layer Observed with Fibre-Optic Distributed Temperature Sensing, Boundary-Layer Meteorology, 154, 189–205, https://doi.org/10.1007/s10546-014-9972-9, http://link.springer.com/10.1007/s10546-014-9972-9, 2015.

RE: **https://doi.org/10.5194/amt-2022-208-RC2**

**Original referee comments in blue, with author response in black.**

This manuscript describes a new modular box enclosure called ISE-CUBE that can be used to deploy water vapor isotopic analyzers and water vapor isotopic cold-trap systems in the field under extreme cold-weather conditions. The manuscript provides a short description of the enclosure and subsequently evaluates the isotopic analyzer's housekeeping variables from a two-week winter deployment in Svalbard. The housekeeping data suggest the analyzer is able to maintain satisfactory ranges for its Data Acquisition System temperature, its cavity temperature and pressure, and its warm box temperature. The analyzer's water isotopic measurement precision in the field is also comparable to its measurement precision while sampling calibration gas in a laboratory setting.

The manuscript also describes an optional "profiling module" for ISE-CUBE, consisting of a tripod with an articulating measurement arm, that can be used to position a heated inlet line for the isotopic analyzer anywhere from 4 to 205 cm above the ground surface. A 90-minute window of data is presented that shows water vapor concentrations and isotope ratios from six height levels within the articulating arm's 2-m range. The paper argues that the articulating arm provides a means to resolve and study the water and isotopic gradients closest to the surface, although doing so requires repositioning the inlet height via a manual pulley every few minutes.

**Comments**

Where this paper really advances our measurement abilities is in the design and presentation of the box enclosure for the isotopic analyzer and cold trap; yet most of these details are in the Supplemental material instead of in the main paper. I would recommend revisiting Figures 1-2 and using these to convey specific details about the box connections and tubing materials, something more akin to the Connectors Template in the SI. As currently presented, Figure 1a is simply too dark to make out details, and Figure 1b requires more detail and explanation. For example, what are the "power out" and "data" ports used for? Where are the fan inlets and exhaust ports? Where do the boxes connect to one another? Is the CRDS inlet unheated after the check valve? Which lines are PTFE and which SS? In addition, the main text mentions components such as an "adapter," an "exterior inlet bulkhead," "incoming ventilation tubing," and "manifold tubing." Can these be labeled on Figs 1-2? The list of components in Appendix A is fantastic. Consider also a corresponding diagram (again, like the Connectors Template) that shows where all these components go and telling readers how many of each part are required to replicate the system. Another way to think about this: what would a purchase list look like?

 As mentioned during the discussion, the focus of the manuscript has shifted towards the system as a profiling platform. We have updated Figures 1 and 2 (now Figure 3), to include annotations. Details on Figures 1-3 are found in the text of Section 2. We've also assembled detailed lists that describe the components necessary for each module, as well as cost estimates, in the Supplemental Material (Section 1.1)

It would be helpful if the manuscript discussed the relevance of the ISE-CUBE enclosure to the wider measurement community. Much of the manuscript is specific to the deployment of a Picarro CRDS water vapor isotopic analyzer. Would ISE-CUBE work for other types of isotopic analyzers? If the enclosure is specific to the size and shape of the Picarro systems, could ISE-CUBE work for other gas-phase Picarro analyzers? Moreover, based on the short two-week deployment in Svalbard, is there any sense whether ISE-CUBE could last for longer periods for unattended measurements?

We now discuss these points in Section 5 Discussion

On a related note, the Data Processing section (Sect. 3.2, including Table 1) presents ISE-CUBE as producing three data streams generally, but these three streams are specific to the way the modular system was set up for testing during ISLAS2020. It would be helpful if the paper distinguished more carefully which aspects of the design are generic and applicable broadly vs. specific to the test case configuration.

Section 3.2 Data processing has been rephrased to distinguish between primary and expansion data modules.

The enclosure is presented as novel, in part, for minimizing disturbance to the environmental flow, but I think this claim might be overreaching, since most ground-based installations are designed to minimize flow disturbance (e.g. flux towers). The real draw of the enclosure in my mind is the ability to deploy a water vapor isotopic analyzer in an environment with minimal infrastructure support (e.g. nothing more than a power drop) and/or to reduce the length of inlet lines and thus measurement hysteresis.

We now discuss the novelty of the ISE-CUBE stack alongside the profiling arm for profiling purposes in Section. 5 Discussion. In this same section, we discuss the individual value of the ISE-CUBE stack for deployments (similar to the comment two above).

To evaluate ISE-CUBE, the water vapor isotopic analyzer's performance in the field is compared to its performance in the laboratory. The intention is to compare two distinct environmental settings. However, there is another relevant difference that needs to be communicated more transparently: in the laboratory, the analyzer samples reference gas continuously, whereas in the field, the analyzer is measuring real variability related to the environment. I would not be surprised if this difference in sampled air causes the differences in humidity-binned standard deviations presented in Fig. 10 or results in the differences in spectral-fit residuals (RS) presented in Fig. 8. The paper concludes that the field data are "marginally less precise," but, again, I wonder if this is not just a reflection of the environmental air. Would one reach the same conclusion if the analyzer were measuring reference gas while deployed as part of ISE-CUBE in the Arctic?

We now conclude Section 4.2 Measurement quality of water vapor isotopes with (L.370) "In summary, the field deployment exhibits consistently higher variability for isotopic measurements, as compared to the optimal measurement conditions in a well-controlled research laboratory" and further mention this possibility in L.372: "but could also be due to the more variable composition of the ambient air used to quantify stability"

While I'm not sure it is necessary for the point the paper is trying to make, it would be awfully interesting to see how the isotopic analyzer and cold trap compare during the ISLAS2020

deployment. Such a comparison could provide some indication of the accuracy of the isotopic analyzer when deployed with ISE-CUBE.

Section 4.4  Cold Trap module performance now includes two analysed samples, compared to CRDS measurements.

Lastly, for Fig. 12, it appears there are environmental data missing during the period highlighted in the text (9:07, onwards). In addition, it would be helpful to know, are these data from the AWS? And can the figure be made larger?

Figure 12 (now replaced by Figure 10) has been refashioned to better represent a profiling operation.

**Overall, the paper is very clearly written; however, a few minor comments on presentation are provided below:**

L 15 - perhaps "components" instead of "compartments"

L.20: " different reservoirs of the hydrological cycle"

L 29-32 is a bit awkward and could be presented more clearly

This sentence has been broken up into smaller pieces in the introduction: L.24-27 pertains to previous structures, and L.28-33 now focus on the inlet lines.

L 39 and elsewhere - "pneumatically" might be the wrong word as this implies compressed air

There really isn't a great single word to replace this; we have now used "sample transmission" or "gas transmission" instead.

L 165 - is there a reference for ISLAS2020?

The data paper for this campaign is in preparation for Earth System Science Data.

L 177 - Does the ISLAS2020 data span 21 Feb to 14 Mar?

It does, though part of that time (29 Feb to 3 Mar) had our instrument installed up at Zeppelin Observatory (472 m ASL). The remaining days were dedicated to calibration and maintenance inside the Marine Laboratory.

L 187 - "reliable" seems like the wrong word for what is intended

We have removed this paragraph.

L 219 requires clarification

L.227: Now reads, "For both isotope species, this standard deviation is similar (or smaller) than the standard deviation typical during any individual calibration."

L 239 and elsewhere - "minutely" means "meticulously." I think the paper intends to say "1 minute"

We have changed all occurrences, and will certainly remember this for future works.

L 375 - since "field" and "laboratory" have specific meanings, I would use "remote observatory" or some other phrase here

This line has been removed in the revised manuscript.

Fig. C1 - caption says observatory margins are for the same "time" but not the same dates, right? Perhaps clarify.

The Figure C1 caption now reads "Black shading denotes spread between 2nd and 98th percentiles of Warm Box temperatures during laboratory benchmark. Gray shading indicates the same, for the deployment period in the Zeppelin Observatory. Blue lines are brief site visits, lasting on the order of 5 to 10 minutes."

L 521 - says "could be" but should it say "is" or does it really mean "might be comparable" (as in it's unknown)?

L. 532 now reads "Accordingly, recovery time for the TWB would also be longer during the profiling periods, though the magnitude of the TWB dip/spike is independent of site visit type."

Fig. D1 seems to be missing the gray shading mentioned in the caption

This figure (or a version of it) has now moved to become Figure 6. Additionally, we have removed the linear regression line and the associated grey shading.

**RE:** **https://doi.org/10.5194/amt-2022-208-RC3**

**Original referee comments in blue, with author response in black.**

General comments:

I found this to be a clearly written paper, well-structured and systematic. It is a little long for the content so an attempt to shorten some sections would be welcome.

Section 4.1 has been substantially reduced, though Section 4.3 Profiling module performance has been extended due to the manuscript now describing a profiling system.

It provides a detailed account of the design and testing of a well engineered set of enclosures for the isotope analyser and some peripheral equipment. It looks to be quite costly - please provide the cost of the parts to build this. Many users of these instruments would look for low-cost solutions to building enclosures and peripherals and in many cases sensible cost cutting can be made without substantially affecting performance.

Section 1.1 in the Supplemental Material now includes detailed breakdowns of what each module contains, in addition to approximate cost. We also put forward some cost-cutting alternatives that wouldn't affect system performance.

The paper would benefit from a better justification of the need for this design, i.e the advantages of this design compared to other solutions to obtain such data. For example, given that a power source must be nearby (presumably within a few 10's of meters) why couldn't instruments be installed indoors with inlet tubing to the outdoors? Long inlet tubings are routinely used on tall masts with an appropriate pump rate? Are there particular problems regarding lag time? Or memory effects? Or disturbance of the air stratification? Why couldn't an existing mast be used with multiple inlet heights and the use of a manifold? A nearby EC tower is mentioned in the paper.

These points are addressed in the revised Section 1 Introduction as well as in the newly added Section 5 Discussion.

The publication could be seen as premature given the preliminary and incomplete parts (cold trap, pivot arm and standard gas supply module). However, it may be that the authors had limited opportunities to test the device in this remote location.

With the new focus as a profiling system, we assert that this near-surface profiling technique is novel, and warrants writing about.

The paper is mainly an account of environmental measurements (T and P, spectral characteristics etc) of the analytical instrument when placed inside the housing under cold and windy conditions, rather than a more complete account of a test of actual isotopic measurements. Ultimately the most critical isotope test, i.e. a comparison of one or more constant gas supplies both in the field and the laboratory, could not be carried out due to the lack of a working standards gas supply module. Also, there seems to have been a missed opportunity to compare the real time isotope data with isotope data obtained from the cold trap sampling, why wasn't this done (or presented) here?

Sections 4.1.2 and 4.1.3 have been streamlined to better address the temperature and pressure stability of the measurement cavity, as well as the spectroscopic diagnostics. Section 4.2 now also better quantifies the measurement quality of the system. L.468 in Section 5 also

now states "Currently, the integrated Cold trap is unsuitable to be used for calibrating CRDS measurements."

Given the anticipated degradation of isotope ratio precision at low H2O mixing ratios (typical of polar regions), and the possible instrument drift due to the changing environmental conditions, it would be important to check standard gasses a regular interval (likely several times daily) in an actual measuring campaign, hence the need for a standard gas supply device. As mentioned, this has not been demonstrated in the current manuscript.

 In Section 5 Discussion, we discuss the potential for the system to be used in longer deployments, in the context that a calibration unit is necessary for this.

A more thorough explanation of the stratified air column data obtained would benefit the paper as it would demonstrate the useful application of the enclosure and pivot arm. The need for an operator to use the pivot arm seems to risk disturbing the stratified air column, depending on wind direction (creating turbulence)

 Figure 10 now shows isotope profiles from a period on 9 March. We also mention the benefit that an automated Profiling module would bring on L.511.

Specific comments:

P1 L12: I don't think you can claim it would be satisfactory in all environments, e.g. in the tropics, overheating, condensation of humidity may be a different challenge, in environments with high day – night temperature contrast drift may be problematic

 L.9: We now only mention Arctic environments.

P2 L41: What are the conventional approaches?

 We have rephrased this part of the Introduction to make it clear that previous in-situ measurements rely on controlled environments.

P2 L49: But you have a nearby power source so why not house the instrument there?

 The nearby power source is the EC mast, which does not have any adequate housing for the Picarro.

P2 L52: But long inlet lines (fluorinated plastics) are routinely used in tall towers

 We now state the motivation behind using short inlet lines in the Introduction (L.35-38).

P3 L64: Is it pneumatic? Maybe just gas or airtight connectors?

 The term "pneumatic" has been removed throughout the manuscript, being replaced by "gas" or "sample transmission"

P4 L85: Isn't a lower flow rate preferable to increase precision in dry air? Especially since there's a separate high-flow pump to deliver the air sample close to the inlet?

 L.480-484 now put forward this potential.

P5 L106: So why wasn't this done when there was no available standard gas module available?

 This sentence has been removed, and we now discuss the suitability of the Cold Trap as a calibration method in Section 5 Discussion.

P6 L144: Does the presence of an operator disturb the stratification?

As mentioned above, we now also discuss the benefit that an automated Profiling module would bring on L.511, in terms of further limiting any potential disruption to the sampling site.

P7 L161: This would have been the most complete test of the system

In its absence, we maintain that the analyser operational/measurement diagnostics put forward provide adequate proof of operation, especially in conjunction with our uncertainty quantification put forward in Section 4.2.

P10 L211: Please state if these are liquid or vapour values

We specify that these are liquid standards delivered by the Picarro SDM system on L.218

P10 L220: This sentence is unclear – does it mean that measurements were within +/- 1 sigma?

Section 3.3 has been revised to better clarify the calibration procedure.

P10 L229: Not sure this is correct, there are numerous field applications documented on the Picarro web site.

This sentence has been rephrased to better emphasize the analyzer's optimal operating environment (L.236).

P11 L247: Don't think 'minutely' is a word (?)

Removed occurrences of "minutely" from the manuscript

P11 L259: Please specify where DAS is measured

L.263: We first use the Data Aquistion System (DAS) temperature ($T_{DAS}$) measured inside the analyzer housing as a proxy of the overall temperature and condition of the analyzer.

P16 L355: As mentioned above, wouldn't the normal low flow setup have been preferable?

L.480-484 now put forward this potential.

P22 L454: add 'for d18) and dD, respectively

Section 4.5 Example of a profiling operation has changed substantially, including this line.

---

## Author Response (AR2)

**Original referee comments in blue, with author response in black.**

1) L. 4: not so clear to me what is meant by "which can have a vertical structure", do you mean the fluxes or the atmosphere.

Rephrased to: "However, gathering observations in harsh environments still poses challenging, particularly in regard to observing the small-scale exchanges taking place between surface and atmosphere. It is especially important to resolve the vertical structure of these processes."

2) L. 5: "lowermost level of the surface layer": do you mean a layer or a specific level? Maybe make this clear by mentioning the first 2 m above the surface, which is what you designed your system for.

Rephrased to: "We have designed the ISE-CUBE system as a modular CRDS deployment system for profiling stable water isotopes in the surface layer, specifically the lowermost 2 m above the surface."

3) L. 25: "structures" -> infrastructure?

Yes, infrastructure is a more accurate term. Changed accordingly

4) L. 28: robustness of the sampled gradient?

Changed accordingly

5) L. 44: Arctic with capital A

Changed accordingly

6) L. 54: induce minimal flow disturbance

Changed to: "The entire system should also induce minimal disturbance to the flow around the measurement site."

7) L. 71: add a manufacturer and a serial number for the check valve for completeness, since you gave this information for all other components (I know that you have it in the table in the Appendix, but for consistency with the rest of the text, I would add it).

Added accordingly

8) L. 75: remove one "the"

Changed accordingly

9) L. 86: remove "now" this is not the first instrument with higher flow rates

Changed to: "... is a custom modification of the standard L2130-i, which enables higher flow rates, similar to the analyzer described in …"

10) L. 99: does Inlet really need a capital I (here and elsewhere)

Changed accordingly, except in cases where a reference is made to text in a figure

11) L. 106: "assign a spatial dimension to these measurements" sounds a bit confusing. How about something along the lines of "We will next describe a profiling system, that enables high-resolution vertical profiling of the lowermost atmosphere", or similar. But make clear it is about sampling vertical gradients.

Changed to: "We will next describe the Profiling module, which enables high-resolution vertical profiling of the lowermost levels of the surface layer."

12) L. 118: "The height of the inlet..."

Changed accordingly

13) L. 187: The last sentence is a bit confusing, your statistical analysis does not allow you to quantify the extremeness of the full period. I would simply state: "Over the time period of the deployment, we encountered several episodes of extreme cold conditions", or something along this line of thought.

Rephrased to: "Additionally, Dahlke et al. (2022) have identified Feb and Mar 2020 as having some of the strongest marine cold-air outbreaks of the last 42 years. Overall, with several episodes of extreme cold, our deployment was a formidable testing ground for the system."

14) L. 213: "to be compared" sounds a bit too qualitative for my taste. How about writing what you actually do, namely you normalise your data to the VSMOW-SLAP scale? Also you should mention what the primary standard is somewhere because you mention that you calibrate with secondary standards, which can be confusing for an unexperienced reader.

Changed to: "Isotopic measurements are calibrated on the VSMOW2-SLAP2 (Vienna Mean Ocean Water 2 - Standard Light Antarctic Precipitation 2) scale, composed of international, primary standards. The use of the scale allows for the relative ratios of heavy to light isotopes ($R_{sample}$) measured in CRDS analyzers to be normalized to a standard ($R_{standard}$), as described in Eq. 1 (Craig, 1961).

…

The value of the resulting $\delta^i$ (with i representing one of the heavy SWIs) is expressed in permil (per thousand, ‰). For our calibrations, we employed two secondary standards, whose isotopic values on the VSMOW2-SLAP2 scale have been established in the laboratory."

15) L. 237: mention the 20° already in the second sentence: well-controlled laboratory setting at room temperature of approximately 20°C. Remove "was operating with ambient room temperatures of approx 20°C" below.

Rephrased to: "The first represents the optimal operating environment for the analyzer, a well-controlled laboratory setting with ambient room temperatures of approximately 20 °C. This first period runs from June–July 2020 when the same analyzer was used at FARLAB, University of Bergen, Norway. During this time, the analyzer was routinely sampling standard vapor with mixing ratios down to 0.155 g/kg, comparable to humidity minimums encountered in the field."

16) L. 239: Merge with the next sentence: "During this time the analyser was routinely sampling standard vapour with mixing ratios down to ... "

Rephrased (see comment above)

17) L. 240: The second period,

Changed accordingly

18) L. 257: I think referring to the different results sections in this first paragraphs would be more intuitive than referring back to the methods section.

Agreed. The paragraph now reads: "We now detail how the field conditions influenced analyzer performance and thus data quality, using the laboratory and observatory periods as performance benchmarks. Thereby, we focus first on temperature and pressure conditions of the analyzer (Sect. 4.1), before evaluating the impact of field conditions on the water isotope measurements (Sect. 4.2). Then we detail the performance of the profiling module, especially the capability of the module to deliver sample to the analyzer for the purpose of resolving vertical profiles (Sect. 4.3). Finally, the performance of the cold trap expansion module is briefly presented (Sect. 4.4)."

Removed $T_{das}$ on L.82, and changed variables to italics

20) Very nice shorting of Section 4.1, the information is now nicely condensed and this part is interesting to read.
Thank you. The comments of the referees in the previous round really helped us pin-point the utility of this section.

21) L. 284 and elsewhere: not sure the correlation value of 0.000 is needed, no correlation is clear enough.
Removed accordingly

22) The y-axes of Fig. 7 are labelled in a slightly disturbing way: it should be sigma for all of them, right? I find d18O,sigma confusing.
Rearranged labels accordingly. Now of the form: $\sigma$, δ¹⁸O (‰)

23) Fig. 7 caption: Differences between bin means from field and laboratory…
Changed accordingly

24) L. 368: this corresponds to a variability increase of 30% in the field compared to the laboratory benchmark.
Rephrased to "Averaged across humidity bins, this corresponds to a variability increase of 30 % in the field compared to the laboratory benchmark, though the largest relative increase occurs at the higher humidities."

25) L. 380: remove one "the"
Changed accordingly

26) L. 426: I find it surprising (and unlikely) that the dD should be so strongly affected by non-equilibrium fractionation effects, while d18O is not... Could it be that the difference comes from the normalisation of the liquid measurements (bias in the depleted standard used), or due to some interaction between the sampled vapour and the tubing (which is likely stronger for dD, but I wouldn't call that a "kinetic effect").
We've removed specific reference to kinetic, but keep mention of fractionation effects in general. The very low temperatures of the collection vial have an impact on the difference between fractionation factors of dD and d18O, which can manifest during incomplete freeze-out. Additionally, we also emphasize that sample loss is a major factor in the comparison disagreement. We have also now included a caption for Table 4.

 Rephrased to: "This is likely due to a combination of deficiencies involving sample collection, inconsistent flow regulation, and possibly fractionation effects at the low temperatures in the glass sample vial. These fractionation effects might arise from incomplete freeze out of the vapor, induced by insufficient thermal stability of the glass collection vial. We additionally observed substantial ice crystal formation in the neck of the collection vials, which inhibited and decreased flow during multiple collection periods. This ice formation also compromised sample recovery during vial exchange, causing frozen sample to fall out of the vial during collection. Therefore, the disagreement between sample and analyzer measurements is not unexpected, especially as the collected sample would no longer correspond with the integrated time period being compared with."

27) Very nice that the authors added a comparison between the on-line vapour measurements from the CRDS system and the cold trap sample.

Thank you. We believe it was worth the effort to support our qualitative assessment with quantitative numbers.

28) Fig. 10 and L. 447: I wondered why some of the higher elevation measurement points have such a small error bar in the dexcess. Shouldn't these error bars also represent the precision of the measurements that were quantified in Fig. 7 to be of about +/-2 permil in dexcess at 0.8 g/kg (3 m height). I think the error bars should represent total uncertainty, even more so since this is how the authors argue at L. 375-377 they would proceed to use their results of Section 4.2.

As we utilize the measurement means to obtain the gradients, we believe that it is more relevant that the errorbars show the uncertainty in that mean. We have now included a variability scale for each profile, denoting a representative standard deviation of any particular height. Sect. 4.5 and Fig. 10 have been revised that reflect this change.

We have also corrected L.375-377, as the increased uncertainty is moreso inherent to our measurement system, rather than something that needs to be added to it, which is how L.375 reads now. This sentence (L.380) has been changed to: "Nonetheless, we will show that though this decreased precision is inherent to the observations, it does not hinder useful measurements, in particular since the measurement precision is quantified."

29) Excellent analysis in Section 4.5! Very nice to see!

Thank you. Just as with shortening Section 4.1, the referees in the previous round highlighted the necessity to expand on the performance of the profiling arm.

30) L. 456: measuring,

I don't think a comma is necessary here as "being able to select heights of interest while measuring" is the subject of the sentence

31) L. 465: context in which, they are taking place.

Changed accordingly

32) L. 466: I think it would be very important to mention that fixed height profiling systems with only few unadjustable inlet heights still have the advantage of allowing to sample automatically over much longer time periods, thereby obtaining a different statistics than with the deployment of a profiling system like the ISE-CUBE. To me the ISE-CUBE is by design operating punctually in a campaign-based setting and it needs an operator.

Changed to: "Finally, it would be quite possible to deploy alongside a tower with fixed height inlets, as these towers have the advantage for automated sampling over much longer time periods."

33) L. 470: could be used without…

Changed accordingly

34) L. 471: I don't understand "a point of diminishing returns"

Rephrased to: "While the Analyzer and Pump modules could be used without the Profiling module for an extended deployment, at some duration the benefits of a more conventional enclosure would begin to warrant additional effort during installation.  A larger enclosure (such as a pre-existing building) offers a level of security that the ISE-CUBEs cannot provide for long-term measurement efforts… "

35) L. 480: I don't really understand the use of this last paragraph. I think it is very unlikely the case that the quality of the measurements would be substantially improved in low flow mode and it would require longer sampling times at a given height.

This point was brought up during the interactive discussion, and we thought it relevant to include it as a possibility.

36) L. 484: most rapid -> fast
Replaced accordingly

37) L. 490: replace in the system by ISE-CUBE
Replaced accordingly

38) L. 498: Due to the high vertical resolution in the profiles, the observed gradients are robust.
Changed accordingly

39) L. 509: yes but unlikely optimised for high flow-rates... isn't this the major challenge here?

The calibration device described in Leroy-Dos Santos (2021) provides flow up to 0.6 LPM, which would be enough to calibrate the analyzer, even when it is in high flow mode. The ~10 LPM flow rate discussed in Sect 4.3 pertains only to inlet characterization (Profiling module performance), not analyzer calibration.

40) Consider moving part of the appendix figures and text into supplements (which would be accessible in separate PDFs and would avoid charging the paper PDF unnecessarily. Of course Appendix A should stay an appendix.

We have moved Appendices B and C into the Supplement.

References

Craig, H.: Isotopic Variations in Meteoric Waters, Science, 133, 1702–1703, 1961.

Dahlke, S., Solbès, A., and Maturilli, M.: Cold Air Outbreaks in Fram Strait: Climatology, Trends, and Observations During an Extreme Season in 2020, Journal of Geophysical Research: Atmospheres, 127, 1–18, https://doi.org/10.1029/2021JD035741, https://onlinelibrary.wiley.com/doi/10.1029/2021JD035741, 2022

Leroy-Dos Santos, C., Casado, M., Prié, F., Jossoud, O., Kerstel, E., Farradèche, M., Kassi, S., Fourré, E., and Landais, A.: A dedicated robust instrument for water vapor generation at low humidity for use with a laser water isotope analyzer in cold and dry polar regions, Atmospheric Measurement Techniques, 14, 2907–2918, https://doi.org/10.5194/amt-14-2907-2021, https://amt.copernicus.org/articles/14/2907/2021/, 2021.